# Hermetic hydrovoltaic cell sustained by internal water circulation

Renxuan Yuan[1,2,3], Huizeng Li [1,3] ✉, Zhipeng Zhao [1,2,3], An Li [1], Luanluan Xue[1,2], Kaixuan Li[1], Xiao Deng[1,2], Xinye Yu[1,2], Rujun Li[1,2], Quan Liu[1,2] & Yanlin Song [1,2] ✉

Numerous efforts have been devoted to harvesting sustainable energy from environment. Among the promising renewable resources, ambient heat exhibits attractive prospects due to its ubiquity and inexhaustibility, and has been converted into electricity through water evaporation-induced hydrovoltaic approaches. However, current hydrovoltaic approaches function only in low-humidity environments and continuously consume water. Herein, we fabricate a hermetic hydrovoltaic cell (HHC) to harvest ambient heat, and have fully addressed the limitations posed by environmental conditions. Meanwhile, for the first time we develop an internal circulation hydrovoltaic mechanism. Taking advantage of the heterogeneous wicking bilayer structure, we verify that inside the hermetic cell, the ambient temperature fluctuation-induced evaporation and further the water circulation can persist, which sustains the hydrovoltaic effect to convert ambient heat into electricity. More importantly, the hermetic design enables the cell to work continuously and reliably for 160 h with negligible water consumption, unaffected by external influences such as wind and light, making it an excellent candidate for extreme situations such as water-scarce deserts, highly humid tropical rain forests, rainy days, and dark underground engineering. These findings provide an easily accessible and widely applicable route for stably harnessing renewable energy, and more notably, offer a novel paradigm toward leveraging low-grade ambient heat energy via circulation design.

The over-reliance on fossil fuels comes to serious concerns such as energy crisis, environmental pollution, and global warming. Recent advances on energy harvesting technology have proposed approaches to achieve sustainable energy from environment, for example, thermal cells[1,2], solar cells[3,4], triboelectric nanogenerators[5,6], moisture energy generators[7,8] and osmotic power generators[9,10]. Particularly, it has been proved that water flow and evaporation on nanomaterials can induce potential and convert ambient heat into electricity, known as hydrovoltaic electricity generation[11–19]. However, the application scenarios of

hydrovoltaic generators have been limited to specific environmental conditions, for example, appropriate humidity[11,20], sufficient illumination[21–23], and rapid evaporation[24–26]. Meanwhile, these devices are susceptible to external disturbances that may cause unstable output.

In this work, we fabricate a hermetic hydrovoltaic cell (HHC) and achieve long-term stable energy harvesting from ambient heat without water consumption and external interference. We propose an internal circulation hydrovoltaic effect. Benefiting from the temperature

[1]Key Laboratory of Green Printing, CAS Research/Education Center for Excellence in Molecular Sciences, Beijing National Laboratory for Molecular Science, Institute of Chemistry, Chinese Academy of Sciences, 100190 Beijing, China. [2]University of Chinese Academy of Sciences, 100049 Beijing, China. [3]These authors contributed equally: Renxuan Yuan, Huizeng Li, Zhipeng Zhao. ✉e-mail: lihz@iccas.ac.cn; ylsong@iccas.ac.cn

gradient induced by ambient temperature fluctuation, the heterogeneous wicking bilayer structure can sustain evaporation and upward capillary flow inside the hermetic cell, thereby forming water circulation. Assisted by these capillary behaviors, hydrovoltaic effect occurs when water flows on the charged surfaces of the bilayer and converts ambient heat into electricity. The HHC can achieve stable output for 160 h, and will not be affected by external factors such as ambient light, temperature, and wind. Meanwhile, profited from the water circulation process, there is no matter exchange of the cell with the outside. Thus, water consumption can be suppressed during the energy generation process, which further extends applications in water-scarce regions. It is anticipated that the universal applicability, the sable output, and the non-consumption make the HHC and this sustainable water circulation strategy efficient for a wide variety of situations.

## Results

### Design and basic performance of the HHC

The scheme of the HHC is demonstrated in Fig. 1a. The cell has a hermetic structure, consisting of a sealed container and an electricity generation unit erected inside the container. A water layer exists inside the container to immerse the bottom of the electricity generation unit. Two inert electrodes are designed on the cover of the container for electricity output. The electricity generation unit is composed of a polyethyleneimine (PEI) modified carbon black (CB) layer and a layer of tissue paper, forming a heterogeneous wicking bilayer on the quartz plate under the effect of capillary force (Fig. S1). To demonstrate the hermetic performance of the HHC, we monitor the mass change of the cell during long-term energy harvesting. As shown in Fig. 1b, the negligible mass change during the 80 h testing verifies that the HHC has excellent airtight property, and the water escape is fully impeded. Under this condition, we demonstrate that the HHC can continuously generate electricity with an open-circuit voltage of 160 mV and a short-circuit current of 200 nA, as the performance of the HHC tested

uninterruptedly for 150 h in Fig. 1c. Note that the inset short-circuit current in Fig. 1c is measured after the long-term testing of voltage, and a long-term test of current is provided in Fig. S2. Meanwhile, the relative humidity (RH) inside the HHC is measured using a commercial hygrometer, maintaining at its maximum value (100%) during the long-term testing (the red curve in Fig. 1c), which is a proof of the notable hermetic character of the HHC. The results are counterintuitive that unlike reported hydrovoltaic strategies depending on continuous water evaporation, our prepared HHC can achieve continuous energy harvesting with zero water consumption. Besides, the hermetic structure of the cell can provide shielding for the cell to eliminate external influences such as wind and humidity. These merits are particularly important and useful for extreme situations such as water-scarce deserts, highly humid tropical rain forests, rainy days, and dark underground engineering. Figure 1d shows the comparison among typical researches and this work (detailed information in Supplementary Table S1), in which the HHC demonstrates superiorities in key parameters, including the energy conservation efficiency, applicable environments (working RH), and the output stability depending on typical environmental resources, usually sunlight. Although the HHC exhibits a moderate open-circuit voltage, considering its zero-water consumption, it still possesses the highest energy conservation efficiency (power generation per unit of evaporation, calculation details shown in Table S1).

### The mechanism of internal circulation hydrovoltaic effect

The HHC can generate electricity steadily and continuously, without matter consumption and dependence on specific environmental conditions, which remarkably differs from the reported renewable energy harvesting strategies[27–32]. To reveal the electricity generation mechanism of the HHC, we investigate the bilayer structure of the HHC (Fig. 2a). As the sectional scanning electron microscope (SEM) images shown, the tissue layer consists of crisscross cellulose fibers with diameters of micron-scale and exhibits a vigorous wicking effect (Fig. S3).

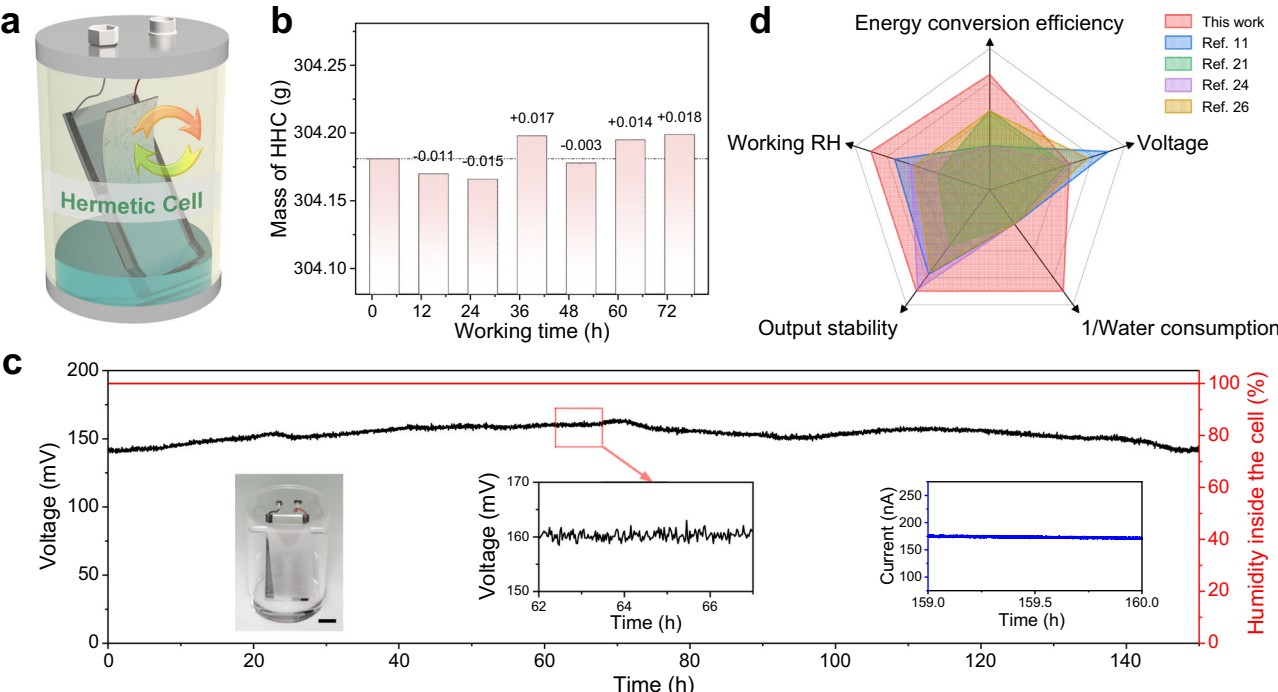

**Fig. 1 | Structure and long-term output performance of the HHC. a** Schematic diagram of the HHC. **b** The mass change of the HHC in the long-term test. The dot dash line is auxiliary line at the initial weight (data collected from the same device). **c** The long-term output performance of the HHC for 160 h. The insets are the optical image of the HHC, the magnification of the open-circuit voltage and the short-circuit current. Scale bar, 2 cm. **d** Comparison on key parameters of hydrovoltaic electricity generators. The collection and evaluation of metrics in Fig. 1d are provided in Supplementary Table S1 and Source Data file. Source data are provided as a Source Data file.

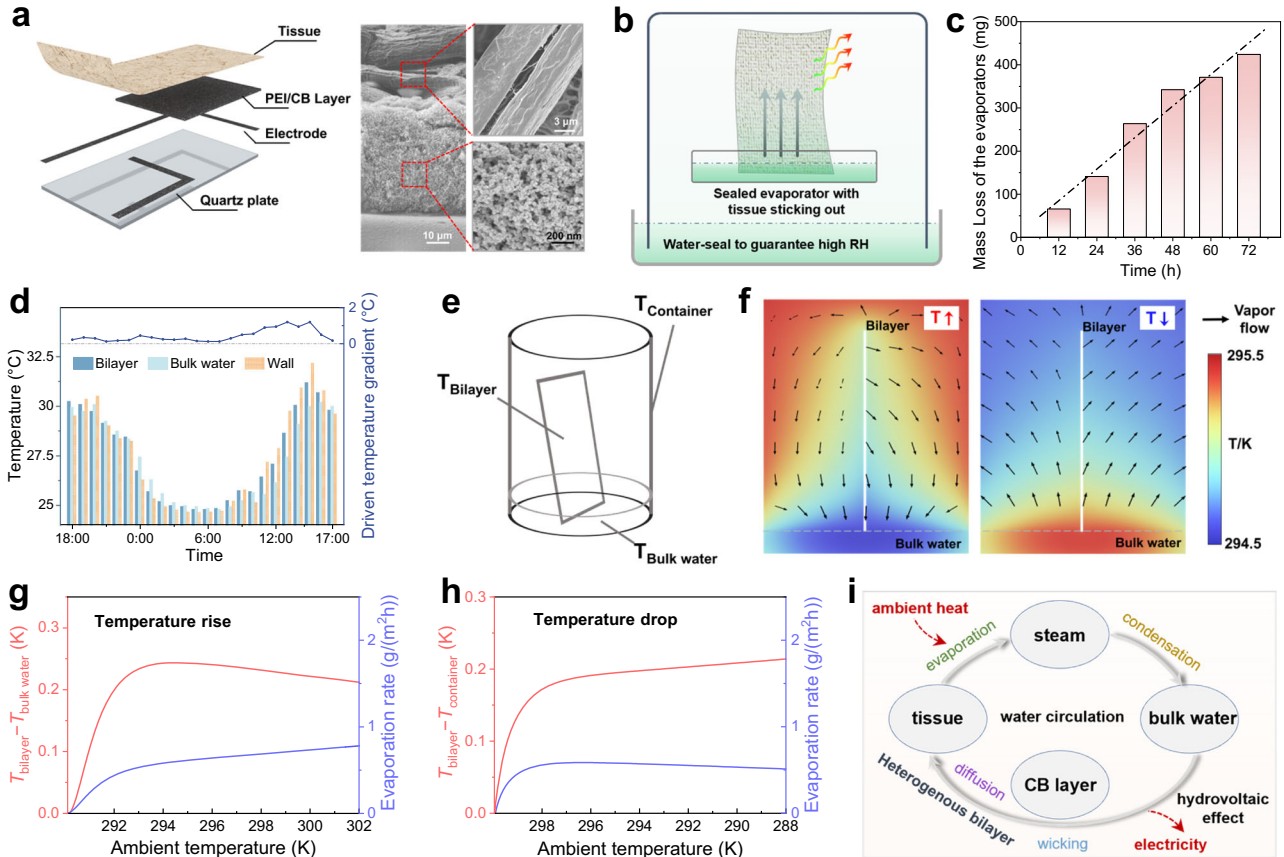

**Fig. 2 | Mechanism of the continuous electricity generation. a** Scheme of the electricity generation unit and the SEM images of the cross section of the bilayer structure (middle), the PEI-modified porous CB layer (bottom-right) and the tissue fibers (top-right). **b** The scheme of the experimental set to verify the continuous capillary evaporation inside a hermetic container. **c** Mass loss of the evaporators for 72 h monitor (*n* = 1). The dot dash line is the linear fitting of the data. **d** The temperature gradients in the hermetic container in 24 h. **e** Scheme of the temperature gradient in the water transport model. **f** The simulated heatmaps of the temperature gradients and moisture diffusion orientation by sectional view. **g**, **h** The dependence of the evaporation rates on the driving temperature differences at varying ambient temperatures. Here the system has a uniform temperature distribution at the beginning, and then temperature gradients emerge with changing ambient temperature. **i** Matter cycle and energy conversion inside the HHC. Source data are provided as a Source Data file.

Between the tissue and the quartz there is a CB layer composed of 20-30 nm CB particles that form a porous microframework (Fig. S4). After PEI modification, the CB layer becomes superhydrophilic (Fig. S5a and S6) with a weak capillary, and forms a heterogeneous wicking structure with the tissue layer. Based on the Gouy-Chapman-Stern theory, the electric double layer (EDL) is formed on the charged surface of the bilayer when contacting with water. For the PEI/CB layer, the Debye length, which indicates the influential range of EDL, exhibits negative correlation with the ion concentration in water[33]. During the upward wicking that is assisted by the tissue layer, water flows through the micro-nano channels between the modified CB nanoparticles. The limited dissociated hydrated hydroxide ions are adsorbed on the surface of the CB nanoparticle, which forms an overlapped EDL[34]. Meanwhile, the positive co-ions, mainly including hydronium protons, selectively pass through the channel to generate electricity through hydrovoltaic effect[11,35–37] (Fig. S7 and S8). For the tissue layer, under the synergetic effect of evaporation and capillary wicking from the heterogeneous bilayer, a bottom-up moisture content gradient along the tissue is formed, leading to nonuniform adsorption of positive ions and further the potential drop difference[38,39]. As a result, a hydrovoltaic potential is created between the wet and dry ends of the tissue due to the surface charge density distribution difference[40–44] (Fig. S9). Detailed discussion is provided in Supplementary Text 3.1. Nevertheless, the revealed mechanism for the hydrovoltaic effect relies on the continuous capillary flows and evaporation, which seems unachievable in such a hermetic container at first glance.

To verify that there indeed exists continuous evaporation in the HHC, we design a confirmatory experiment to explore the water transport in the closed system. A closed evaporator with a water layer at the bottom and a tissue sticking out is placed into a water-sealed container, as the scheme shown in Fig. 2b and details of the experimental design in Fig. S10. It can be rationally hypothesized that the RH inside the container and the water distribution in the tissue achieve dynamic equilibrium in a certain period of time (usually in several hours). A group of evaporators are weighed after taking out from the containers with a 12-h interval, to test whether the evaporation persists after attaining the dynamic equilibrium. The mass losses of the evaporators with different sealing time are summarized in Fig. 2c. It clearly shows that the amount of evaporated water increases linearly with time, indicating the steady evaporation rate on the tissue in the water-sealed container. Therefore, it can be concluded that the tissue-loaded capillary water indeed evaporates continuously in the HHC (Fig. 1b), thus providing upward capillary flow for the hydrovoltaic electricity generation.

In the following, an internal circulation model is developed to reveal the water transport mechanism inside the HHC, which includes wicking flow in the bilayer, evaporation from the tissue[45–47], moisture transport, and condensation (Fig. S11). Under the persistent ambient temperature fluctuations (Fig. S12), the low heat conductivity of humid air and the high heat capacity of water surely induce temperature gradients inside the container[48]. The temperature gradients induced by the ambient temperature fluctuations are recorded in 24 h (Fig. 2d and Fig. S12a). The result indicates that the driven temperature

gradients exist during the whole process. Further, we study the water transport under the internal temperature gradient, considering the temperatures of the bilayer, the bulk water, and the container wall (Fig. 2e). As the simulated results in Fig. 2f, the temperature of the bilayer is higher than that of the bulk water and the container when ambient temperature rises and drops, respectively (detailed information in Supplementary Text 3.2). Meanwhile, the humidity at different locations of the container approaches saturation because of evaporation. Since the saturated vapor pressure depends on temperature, the temperature gradients indicate moisture concentration gradients[49,50]. Therefore, the evaporated moisture diffuses from the bilayer to colder surfaces, as the vapor flow demonstrated in Fig. 2f. Note that the diffusion is the rate-determining step of evaporation. The relationships between the evaporation rate and the temperature gradients at varying ambient temperatures is investigated with numerical methods, as plotted in Fig. 2g and Fig. 2h. It clearly shows that water can evaporate stably from the bilayer over the investigated range with a rate of 0.5 g/(m²h), which well-matches with our experimental result of 0.6 g/(m²h) in Fig. 2c, validating the proposed internal circulation mechanism. It is believed that the diurnal temperature differences influence the general output via evaporation rates.

Further, a simplified model is built to reveal the influential factors on the evaporation rate $\dot{m}$, which is limited by the diffusion rate as $\dot{m} \sim k \frac{DpL}{T^3} \frac{\partial T}{\partial t}$, where $T$ is the ambient temperature, $D$ is the diffusion coefficient of air, $p$ is the saturated vapor pressure, $L$ is the latent heat of vaporization, $\frac{\partial T}{\partial t}$ is the ambient temperature changing rate. $k$ can be regarded as a constant with expression of $k = MC_{p,a}\rho_a x_w / \lambda R^2$, in which $M$ is the molar mass of water, $C_{p,a}$ is the specific heat capacity of air, $\rho_a$ is the density of air, $x_w$ is the radius of the container, $\lambda$ is the heat conductivity of air, and $R$ is the gas constant (detailed analysis in Supplementary Text 3.3). The well agreement between the theoretical analysis, the simulation results, and the experimental data indicates the reasonability of the proposed mechanism (Fig. S13). Therefore, the matter circulation and energy conversion paths involved in the HHC can be summarized in Fig. 2i. This novel cyclic process, which has not been disclosed or utilized, holds significant promise for the broadly applications of sustainable energy. Moreover, the electricity generation by the HHC is non-consumptive and unceasing, which provides a long-term suitable and sustainable strategy to harvest green energy.

The ambient heat, as an inexhaustible but low-grade energy source, is harnessed through the spontaneous and continuous capillary evaporation, and finally outputs in the form of electricity by the hydrovoltaic devices. Based on the model proposed above, it is difficult to directly measure the input ambient heat. Therefore, we propose an alternative efficiency calculation method that focuses on the energy harvesting from the evaporation, by calculating the proportion of generated electricity in the output energy of the vaporization heat and the electricity. An approximate formula is employed: $\eta \approx \frac{VI}{2L\dot{m}A}$, where $\eta$ is the energy conversion efficiency, $V$ and $I$ are the open-circuit voltage and the short-circuit current of the hydrovoltaic devices, $L$ is the latent heat of evaporation, and $A$ is the area of the evaporation region. Though the output performances of the hydrovoltaic devices differ by more than hundredfold, taking the size of the devices into consideration, most of these devices have similar energy conversion efficiencies. Limited by its hermetic structure, the HHC holds the evaporation rates of ~$10^{-3}$ kg/(m²h), which is about 1000 times smaller than reported results. However, it exhibits comparable energy conversion efficiency with the other devices that are open in air (detailed analysis and comparison are provided in Supplementary Text 3.5).

## Influential factors and performance optimization of the HHC

As stated above, one of the major merits of the HHC is the hermetic structure, which avoids interferences from environmental factors such as humidity and wind (Fig. 3a), and facilitates reliable output. However, the temperature and light might affect the evaporation and also the electricity generation performance. For verification, we place the HHC in an incubator and explore the output variation at different temperatures. As concluded in Fig. 3b, the short-circuit current enlarges consistently from 146 nA to 411 nA with the increase of equilibrium temperature from 10 °C to 50 °C, while the open-circuit voltage remains stable around 160 mV throughout the temperature range[51]. This can be explained that the voltage mainly depends on the ion adsorption, which shows positive correlation with the moisture content gradient between the top and bottom of the tissue. In the investigated temperature range, the moisture content at the top of the tissue is significantly lower than that of the bottom with strong capillary effect, thereby generating a steady voltage. Dissimilarly, the temperature rise promotes the water evaporation rate, which accelerates the ion mobility and contributes to the enlargement of the current. Further, the effect of the temperature fluctuation is explored. In order to make the results more obvious, we amplify the temperature fluctuations from a negligible range to a large range of about 50 °C per hour. The temperature increases from 15 °C to 48 °C in 40 min to induce a serious drop-off of RH inside the cell. The open-circuit voltage suddenly goes up by 13 mV, then returns to and remains at 152 mV (Fig. 3c). The escalation on the open-circuit voltage firmly supports the proposed conclusion that the voltage is originated from the unsaturation of the internal RH. This can be explained that the drop-off of RH accelerates the water evaporation and alters the water distribution in electricity generation unit, thus enlarging the potential drop difference. On the other hand, the return of the voltage shortly after the increase is attributed to that the RH restores equilibrium and the moisture content gradient restabilizes. More importantly, the result evidences that the open-circuit voltage can remain stable under ambient temperature fluctuations (Fig. 1c). Also, to prove that it is the continuous ambient temperature fluctuations maintain the evaporation as well as the electricity output, the ambient temperature in an experiment is managed to be nearly constant 0 °C with ice-water bath (Fig. S14). The nearly-zero final open circuit voltage suggests the indispensability of the ambient temperature fluctuations.

In addition, we find that the output of the cell can be promoted with intense light (Fig. 3d). Though no discernible difference emerged between dark and weak light conditions, the open-circuit voltage increases to 197 mV under 1 sun. Obvious growth take place in short-circuit current that increases to 283 nA under 0.5 sun and further to 404 nA under 1 sun. This can be interpreted that the black PEI/CB layer possesses high light absorption capability, thus the localized photo-thermal effect significantly enlarges the moisture content gradient inside the HHC, improves the water evaporation rate, and therefore enhances the electricity generation[52].

Then the influence of the circulation liquid and the type of the electrodes to the performance of the HHC is explored, since diverse liquid and electrodes have been previously employed[18,21,25,53–55]. Taking ethanol, deionized water (DI water), and NaCl solutions of different concentrations as example. As shown in Fig. 3e, the invisible output of ethanol comes from its aprotic properties, since no dissociated ion can adsorb on surface to form the EDL. For the NaCl solutions, on the one hand, increasing the adsorbed salty ions raise the surface charge density, generating larger voltage in the tissue layer. On the other hand, the high ion concentration makes the Debye length narrower than the size of the microchannel in the PEI/CB layer, which hinders the overlap of the EDL and blocks the hydrovoltaic effect in the PEI/CB layer[33,56], according to the Supplementary Text 3.1. As a result, the open-circuit voltage decreases to 91 mV in 0.001 M NaCl solution and further to 75 mV in 0.1 M NaCl solution, as shown in Fig. 3e. Another easily overlooked but important factor is the electrode[26,42,56–58]. To achieve sustainable and environmental-friendly electricity generation, it is significant to avoid the possible consumptive redox reaction on electrodes. As shown in Fig. 3f, inert Au and Pt electrodes show similar output performance, and are comparable with the printed CB

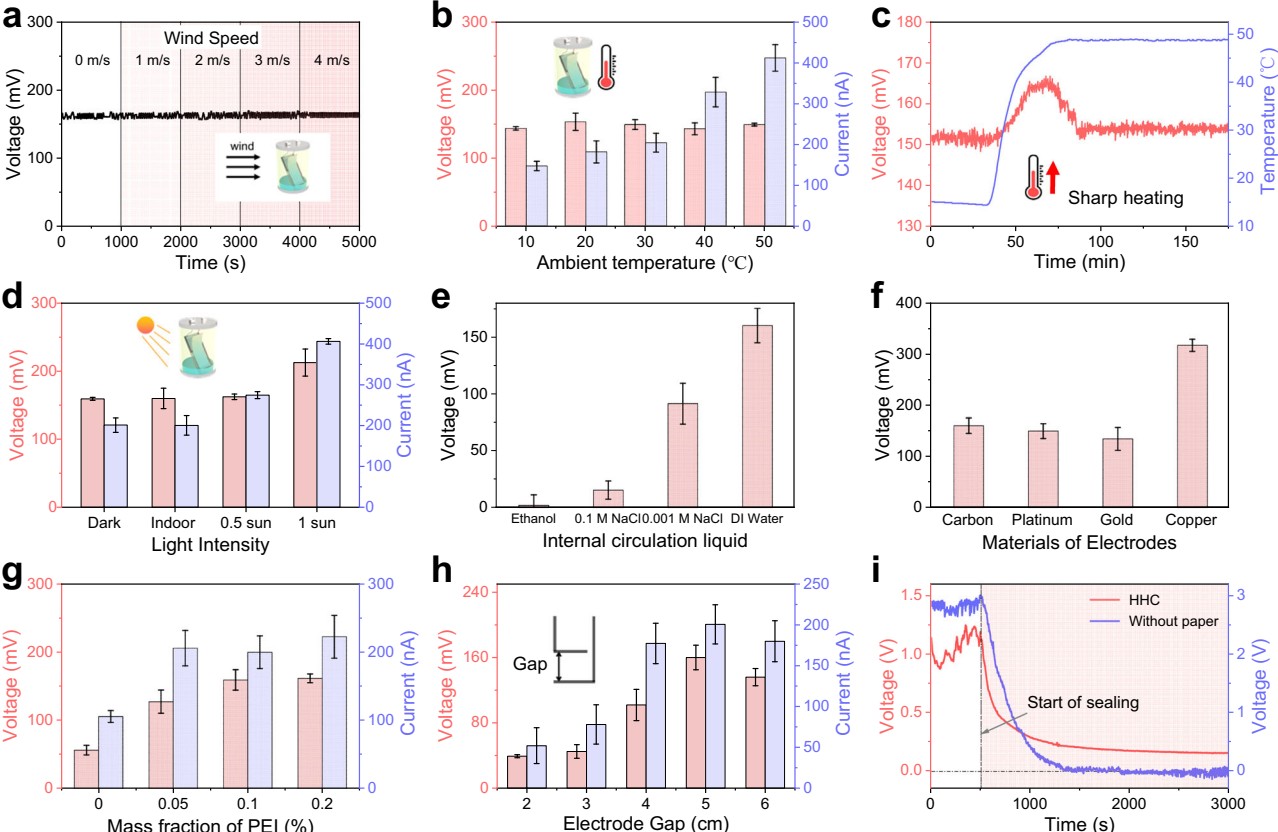

**Fig. 3 | Influential factors of the output performance of the HHC. a** The voltage remains stable at different ambient wind speeds. **b** The voltage stabilizes while the current rise with the temperature increasing. **c** The voltage increases rapidly and then restores with a dramatic change in temperature. **d** The influence of the light intensity. **e–h** The HHC fabrication parameters, including the internal circulation liquid, the electrode types, the PEI concentration, and the electrodes gap to its output performance. **i** The performance comparison of the hermetic cells with and without tissue. For (**b**) and (**d–h**), data are collected from different devices ($n = 3$, error bars represent SD). Source data are provided as a Source Data file.

electrodes. By comparison, the copper electrode, on behalf of active electrodes with redox reaction, indeed achieves higher open-circuit voltage of 314 mV due to the galvanic effect. Considering the sustainability and cost-effectiveness, the printed CB electrode is adopted in our research.

Finally, the influence of the parameters relating to the heterogeneous wicking bilayer is investigated and optimized. For surface properties such as wettability and charging, appropriate functional modifications can vastly enhance the performance of the HHC[13,59]. For example, as the mass fraction of modifying PEI solution increases from 0 to 0.10 %, the open-circuit voltage increases from 57 mV to 160 mV (Fig. 3g). Further increase of the mass fraction of PEI saturates the functional amino groups on the CB surface, so the open-circuit voltage remains unchanged. Besides surface properties, we find that the scale of the electricity generation unit is crucial to the open-circuit voltage. On the one hand, the variation in the unit width shows ignorable influence to the HHC performance[22]. On the other hand, the open-circuit voltage changes obviously at different unit lengths (the electrode gap, Fig. 3h). For explanation, we revisit the water distribution in the heterogeneous wicking bilayer. The increase of the gap length enlarges the area of the thin liquid film in the unit and reshapes the water distribution, which contributes to the formation of the overlapped EDL and the nonuniform ion adsorption by altering the surface charge density, thus boosting the open-circuit voltage. Further extending the gap length results in that the evaporation effect overwhelms the capillary wicking in the bilayer, making the capillary water insufficient to produce a reliable hydrovoltaic effect. The HHC with multiple kinds of tissue paper is also investigated in Fig. S15. To

highlight the necessity of the bilayer to the water circulation and the electricity generation, hermetic cells with and without tissue are fabricated, and the results are summarized in Fig. 3i. The hermetic cell without tissue generates a voltage of 3 V when unsealed, which sharply drops to 0 V in less than 1000 s after being sealed. By compassion, the voltage of the hermetic cell with tissue flatly decreases from 1 V to 0.16 V and keeps stable. When the cell without tissue paper is sealed, the ultrahigh specific surface area of the porous carbon and the high RH induce a water layer on the CB layer, vanishing the water content gradients and further the electricity generation[11] (Fig. S7a). Oppositely, in the cell with tissue, the tissue layer absorbs water from not only the bulk water but also the CB layer, generating the water content gradients induced electricity in the bilayer, which evidences the important role of the tissue paper for continuous electricity generation inside the HHC.

**The scale-up and the application of the HHC**
Furthermore, the output performance of the HHC with different external resistance is probed to demonstrate its application in electricity generation. As shown in Fig. 4a, with the increase of the external resistance from 0.05 to 15 MΩ, the voltage on the resistor increases from 14 mV to 156 mV while the current decreases from 248 nA to 10 nA. Accordingly, the output power is calculated by $P_{OP} = I_{SC}^2 R$, where $P_{OP}$ is the output resistive heat power, $I_{SC}$ is the short-circuit current and $R$ is the external resistance. The power output reaches to the maximum of 8.24 nW with an external resistance of 0.7 MΩ. We find that the voltage and the current on the external resistance can be well fitted as a simplified equivalent circuit by $V_{OP} = \frac{R}{R+r} \times V_{OC}$ and

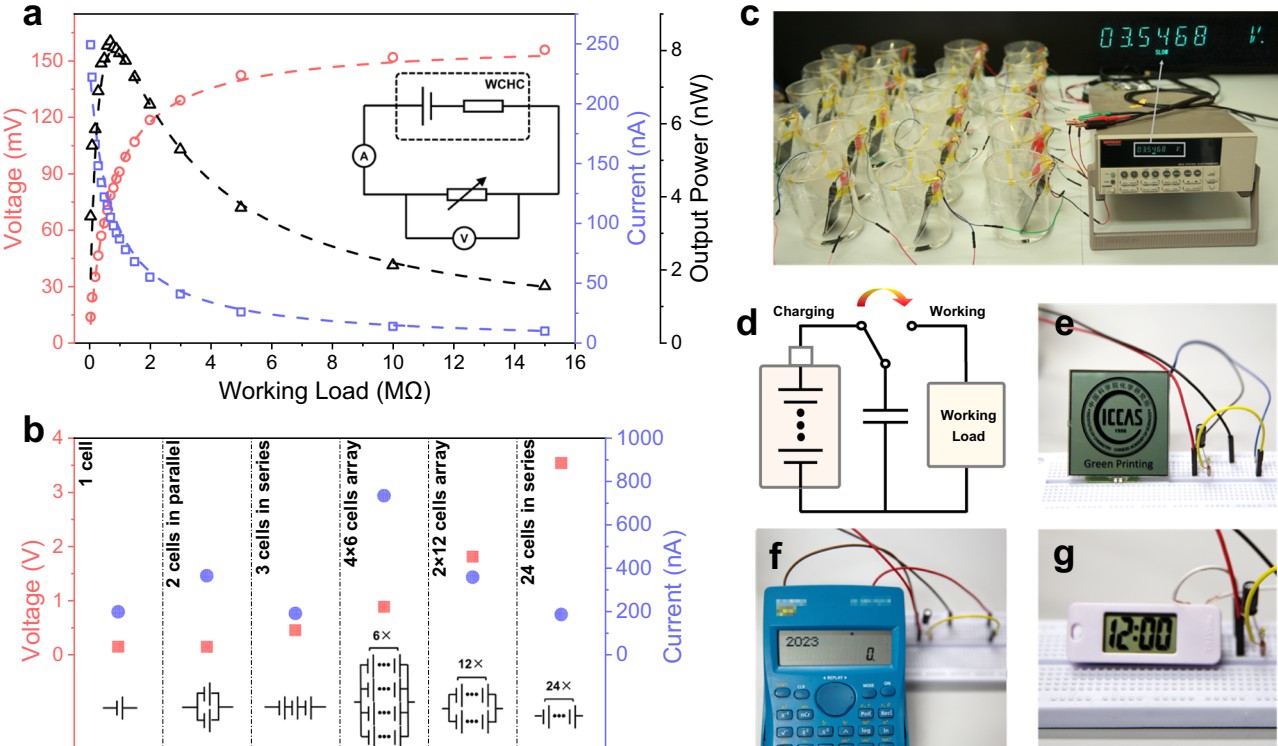

**Fig. 4 | Demonstration of the scalability and the application of the HHC. a** The open-circuit voltage, the short-circuit current and the calculated output power under different working loads. **b** Performance of the HHC array with different connections. **c** The optical image of 24 HHCs connected in series. **d**–**g** Circuit diagram and photographs of powering up electronic devices. Source data are provided as a Source Data file.

$I_{SC} = \frac{V_{OC}}{R+r}$, where $V_{OP}$ is the output voltage on the external resistance, $r$ is the hypothetical internal resistance of the HHC, $V_{OC}$ is the supposed open-circuit voltage of the HHC. Therefore, the HHC can be approximately regarded as an ideal direct current power supply with internal resistance of 0.7 MΩ (see Supplementary Text 3.6). Further, the HHC can be scaled up and be connected in series and parallel configurations to multiply its output[60,61]. The electricity performance of the HHC arrays is summarized in Fig. 4b. Insets show the circuit configuration. As excepted, the open-circuit voltage and the short-circuit current of the arrays enlarge linearly with the amount of the cells connected in arrays. The outcome indicates that the HHC is able to be easily expanded for scalable applications with neglectable fluctuation and zero water consumption. Fig. 4c demonstrates a HHC array with 24 cells connected in series, which reaches an open-circuit voltage of 3.55 V. This HHC array can power up common electronic devices (Supplementary Video 1), such as LCD screens (Fig. 4e), calculators (Fig. 4f) as well as digital clocks (Fig. 4g), with the assistance of a capacitor, as the circuit configuration shown in Fig. 4d.

## Discussion

In conclusion, we demonstrate a hermetic hydrovoltaic cell and realize sustainable electricity generation from the ubiquitous and inexhaustible ambient heat energy, while avoiding water consumption and preventing external disturbances. More importantly, we disclose a previously unperceived internal circulation hydrovoltaic effect, interpreting the energy harvesting capability in the hermetic cell. Combined with experimental, theoretical, and simulative results, it is concluded that the slight fluctuations in ambient temperature and the designed heterogeneous wicking bilayer enable the continuous evaporation and capillary flow inside the hermetic cell, forming water circulation process. This sustains electricity generation by hydrovoltaic effect. More importantly, we demonstrate that the ubiquitous and perpetual temperature fluctuations, which are commonly

considered detrimental or neglected during energy harvesting, allow stable and sustainable power generation through closed-loop matter circulation. We anticipate that the hermetic hydrovoltaic cell and the internal circulation hydrovoltaic effect enables the generation of electricity with low cost, easy accessibility, and wide applicability, which offers an exciting and efficient route for sustainable energy harvesting. Moreover, the constructed water circulation system as well as the associated energy conversion processes would definitely inspire the innovative design of diverse devices.

## Methods

### Materials

Glutaraldehyde (50 wt.% in water) and Polyethyleneimine (PEI; MW ~ 600) were purchased from Shanghai Aladdin Biochemical Technology Co., Ltd. Ethanol was purchased Concord Technology Co., Ltd. Quartz plates (45×80×1 mm³) were purchased from Jiangsu Julin quartz Co., Ltd. The tissue (Xuanzhi) was purchased from Anhui Juyuntang Paper Co., Ltd. Copper Dupont wires, capacitances, as well as other electronic components were commercial products purchased from Shenzhen Kebiwei Semiconductor Co., Ltd.

### Fabrication of the HHC

Quartz plates were cleaned with ethanol and DI water. The CB electrodes were printed by a handmade direct-writing equipment. Typically, the gap between two electrodes was 5 cm. After drying at 80 °C for 10 min, these quartz plates were cleaned again with ethanol. The CB layer was deposited on the quartz plates through flame synthesis method. After ultraviolet radiation (HW-UVX-400, Zhonghe) for 2 h, the plates were immersed in the 0.10 wt.% PEI solution in a 70 °C water bath for 1 h. Then, these PEI modified plates were dipped in 0.05 wt.% glutaraldehyde aqueous solution for 1 h for crosslinking reaction. The tissue slices were assembled on the modified CB layer. Finally, the plate was placed in a sealed container with a water layer of 1 cm at the bottom.

## Characterization

The morphology of the modified CB layer, the tissue, and the CB powders were characterized by scanning electron microscopes (JSM-7500F, JEOL, and Regulus 8230, Hitachi) and transmission electron microscope (HT 7700, Hitachi). All electricity characterization was performed by a Keithley 6514. The relative humidity and the temperature were recorded by a digital thermo-hygrometer (GSP-6, Jingchuang). The water contact angles of the deposited CB layer, as well as its modified surfaces, were characterized by an optical contact angle measurement equipment (DSA100, KRUSS). The zeta potential of the modified CB layer was measured in KCl aqueous solution with pH of 6.94 by a solid surface zeta potential analyzer (SurPASS, Anton Paar).

## Measurement of evaporation in hermetic containers

To verify the capillary evaporation in the hermetic container, we water-sealed the upturned beakers as hermetic containers. Sealed evaporators were fabricated, including a water layer at the bottom, a tissue slice with the lower-end immersed into water and upper-end sticking out the evaporator, and a superhydrophobic holder hanging the tissue above the evaporator (as schemed in Fig. S10). The evaporator was weighed and immediately sealed into the water-sealed container. The evaporators were subsequently taken out from the containers and weighed with a 12-h interval. The data points were linearly fitted (dash line in Fig. 2c). The high R-square of 0.96 suggested a stable evaporation rate in the hermetic containers.

The mass changes of the HHC were monitored with a 12-h interval, and the results were summarized in Fig. 2c. The data points were compared with the base line of zero mass loss, from which the negligible matter exchange with the outside environment was indicated.

## Simulations of environmental conditions

Wind of different speeds was stimulated by an axial flow fan, and was standardized with an anemograph (SW6026, Suwei). The temperature control experiments were performed in an incubator (HWS-150B, Enyi). The temperature curve was recorded by the digital thermo-hygrometer (GSP-6, Jingchuang). Sunlight was provided with a solar simulator (Enlitech, SS-X100R).

## Simulations in COMSOL

COMSOL Multiphysics 6.1 was employed to model the water circulation process in the HHC with numerical methods. With realistic modeling, the conditions of temperature increasing, decreasing and fluctuations were stimulated to cover all situations. Detailed information is provided in Supplementary Text 2.2.

# Data availability

The authors declare that the data supporting the findings of this study are available within the paper and its supplementary information files. Source data are provided as Source Data file. Source data are provided with this paper.

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

## Acknowledgements

The authors thank Prof. Lisbeth Garbrecht Thygesen from University of Copenhagen for her valuable advice. This work is supported by the National Key R&D Program of China (2023YFE0111500 (Y.S.)), the National Natural Science Foundation of China (22272182 (H.L.), 52293473 (H.L.), 51903240 (H.L.), 52321006 (Y.S.), T2394484 (Y.S.)), Beijing National Laboratory for Molecular Sciences (BNLMS-CXXM-202005 (Y.S.)), the Youth Innovation Promotion Association of CAS (2023039 (H.L.)), and the Beijing Nova Program (20230484291 (H.L.) and 20240484643 (H.L.)).

## Author contributions

H.L., Y.S., and R.Y. conceived the project. R.Y., H.L., and Z.Z. performed the experiments. R.Y., H.L., and Y.S. analyzed the data and discussed the results. R.Y., H.L., and Y.S. wrote and proofread the paper. A.L., L.X., K.L., X.D., X.Y., R.L., and Q.L. discussed the result and helped modify the manuscript.

## Competing interests

The authors declare no competing interests.
