## [Transparent Peer Review file · Nature Communications]

Hermetic hydrovoltaic cell sustained by internal water circulation

Corresponding Author: Professor Yanlin Song

Version 0:

Reviewer comments:

Reviewer #1

(Remarks to the Author)

In this manuscript, Yuan et al. presented a hermetic hydrovoltaic cell (HHC) to harvest ambient heat and have fully addressed the limitations posed by environmental conditions. They first proposed a hydrovoltaic mechanism for sustainable electricity harvesting via an internal circulation. Inside the hermetic cell, the natural fluctuation of ambient temperature induced evaporation and further water circulation can persist, persistently converting ambient heat into electricity by the hydrovoltaic effect. The hermetic design enables the cell to work continuously and reliably for one week with negligible water consumption, with no effect from the external environments. While the results are interesting, the authors need to address following issues before the manuscript can be considered for publication.

1. In this manuscript, the abbreviation "WCHC" is not explained.
2. "The electricity generation unit is composed of a polyethyleneimine (PEI) modified carbon black (CB) layer and a layer of tissue paper, forming a heterogeneous wicking bilayer on the quartz plate." How is the tissue paper assembled on the modified CB layer? Will the tissue paper peel off after long-term testing?
3. The hermetic cell without tissue sharply drops to 0 V in less than 1000 s after being sealed. The voltage of the hermetic cell with tissue flatly decreases from 1 V to 0.16 V and keeps stable after being sealed. Why does the voltage of the hermetic cell with tissue keep stable after being sealed? What's the function of the tissue paper? Does the continuous evaporation remain in the hermetic cell without tissue?
4. "a short-circuit current of 200 nA...in Fig. 1c." The figure only shows the short-circuit current from 159 h to 160 h.
5. The water evaporation is induced by the ambient temperature fluctuation. When the ambient temperature is managed to be constant, will the water evaporation and hydrovoltaic phenomenon disappear?
6. The authors are suggested to perform composition analysis for the wicking bilayer before and after long-term energy harvesting to see whether any change in the chemical composition. I'm wondering whether there is some chemical reactions takes place within the container.
7. Since the ambient temperature fluctuation is critical to maintain the electricity harvesting, the authors need to carefully compare the electric signal (e.g., figure 1c) to the variation trend of ambient temperature (e.g., figure 2d). Is there any correlation between the two variables? Of not, what is the reason?
8. In Page 24, "The mass change...in Fig. 2d." The data does not match figure.
9. In Fig. 2g and h, why does the temperature gradient between bulk water and the bilayer decrease at high ambient temperature, while the temperature gradient between the container and the bilayer increases? Does the size of the container influence the performance of HHC?

Reviewer #2

(Remarks to the Author)

The author proposed a hydrovoltaic effect inside the hermetic cell. However, the reviewer is confused about some points in operation mechanism and evaluation as an energy harvesting technology. The reviewer can only judge the manuscript after checking some important experimental results suggested below.

At 100% relative humidity, although there may be a dynamic equilibrium of evaporation and condensation of water, this equilibrium theoretically exists at any point in the environment. It has nothing to do with the material structure. Why can the container exhibit evaporation in one area and condensation in one area?

What is the role of tissue, and how does it play the role?

Whether can the authors provide the relevant experimental data of temperature gradient? What is the causal relationship between temperature gradient and evaporation condensation?

Is it possible that the experimental results in FIG. 2b and c were caused by the fact that tissue and water did not reach the equilibrium of water absorption in the initial stage? So the tissue continues to absorb water resulting in a decrease in the solution? Why doesn't the condensed water return to the original container?

If the energy output of the device comes from the heat energy in the environment, when the entire system is placed in an adiabatic environment, and the electrical output is outside the adiabatic environment, will the temperature inside the device reduced like a refrigerator?

Reviewer #3

(Remarks to the Author)

The authors present a truly unique system of hydrovoltaics that works under conditions that other hydrovoltaics do not; hermetically sealed. This type of system could have a great number of advantages in actual deployment of devices since other technologies would be strongly affected by environmental conditions while these would not.

The methodology and work is sound, however, I think there is a major opportunity missed herein. One question that remains in hydrovoltaics is a measure of the efficiency of thermal energy conversion. With this sealed system the temperature changes induced by evaporation and the resulting energy conversion could possibly be used (almost like a bomb calorimeter) to measure the quantum efficiency of the energy conversion process. I would expect no less from an article in Nature, to be even more impactful.

The authors did not compare their devices to other devices that have been reported and have orders of magnitude higher power outputs. This report is scientifically interesting but it is a bit misleading to report this device as comparable with high performance hydrovoltaics. With respect to the above discussion on efficiency of this device, I would like to see a lengthy discussion of how these sealed devices can not provide comparable power to devices that are open to the environment. Maybe long term one could make that argument but these are really really not powerful, especially considering the size of the device.

There are a few minor issues with spelling and grammar where the manuscript might benefit from another round of editing. Also, I am not sure if it is a sign convention difference or if the authors are using absolute values for current and voltage but these would typically be different signs. Could the authors explain this?

Reviewer #4

(Remarks to the Author)

Version 1:

Reviewer comments:

Reviewer #1

(Remarks to the Author)

The authors have addressed most of the issues I raised in the first round of review. However, several issues remain to be addressed before the manuscript can be accepted for publication.

1. The measurement of short-circuit current from 159 h to 160 h may mislead the readers that the short-circuit current can last for 160 h. It should mention that the short-circuit current was measured immediately after the open-circuit voltage and lasted only for 1 hour.
2. For the comment 9, according to the author's reply, the temperature gradient inside the HHC is determined by the change in heat conductivity of air induced by ambient temperature. Why does the temperature gradient between bulk water and the bilayer ($T_{\text{bilayer}} - T_{\text{bulk_water}}$) increase and then decrease as the temperature rises?
3. The authors are suggested to fix the format errors, such as Page 21, line 433.

Reviewer #2

(Remarks to the Author)

The authors have provided a clearer explanation of the operation mechanism of the hermetic hydrovoltaic cell (HHC) by adding extensive experimental results and related analysis. Although the output performance of HHC is not considered outstanding compared to previously reported works, it does show unique advantages in the range of applications. However, the following comments need to be addressed before the manuscript could be recommended for publication:

1. It is recommended that the authors perform a long term test on the short circuit current of HHC, this data is important to assess the long term output stability of HHC.
2. As shown in Fig.4c, although the electrodes on the quartz plate of HHC are made by printing CB, it seems that they are

connected out from the inside of the container to the outside with copper wires, does this part have an impact on the output performance?

3. In the bilayer structure, can an electrical signal also be generated continuously if the tissue does not completely cover the CB layer? For example, if the tissue only covers the CB layer in the non-bulk water, what is the output performance of the HHC?

4. How does the pore size and thickness of the tissue affect the output performance of HHC?

Version 2:

Reviewer comments:

Reviewer #1

(Remarks to the Author)

The authors have addressed my comments and the manuscript can now be published.

Reviewer #2

(Remarks to the Author)

After thoroughly reviewing the comments and revised manuscript, the author has answered most of the questions. However, I suggest that the author provide efficiency and provide a detailed explanation of the efficiency of this device in the main text, which is very important from an application perspective as this aspect has rarely been addressed in previous publications.

Version 3:

Reviewer comments:

Reviewer #2

(Remarks to the Author)

It is OK for acceptance now.

Reviewer #1 (Remarks to the Author):

In this manuscript, Yuan et al. presented a hermetic hydrovoltaic cell (HHC) to harvest ambient heat and have fully addressed the limitations posed by environmental conditions. They first proposed a hydrovoltaic mechanism for sustainable electricity harvesting via an internal circulation. Inside the hermetic cell, the natural fluctuation of ambient temperature induced evaporation and further water circulation can persist, persistently converting ambient heat into electricity by the hydrovoltaic effect. The hermetic design enables the cell to work continuously and reliably for one week with negligible water consumption, with no effect from the external environments. While the results are interesting, the authors need to address following issues before the manuscript can be considered for publication.

We deeply thank you for the time and effort in thoroughly reading our manuscript. Especially, we are very appreciative of your interests in our work, as well as providing constructive suggestions, particularly on the relation between the evaporation phenomenon and the electricity generation. Encouraged and motivated by these comments, we have carefully modified our manuscript. We believe that the revised manuscript would dispel the concerns.

Comment 1: In this manuscript, the abbreviation “WCHC” is not explained.

Reply 1: We apologized for the typos in the manuscript. All the abbreviations “WCHC” have been replaced by “HHC” in the revised version.

Manuscript modifications:

The typos in the Manuscript are corrected.

Comment 2: “The electricity generation unit is composed of a polyethyleneimine (PEI) modified carbon black (CB) layer and a layer of tissue paper, forming a heterogeneous wicking bilayer on the quartz plate.” How is the tissue paper assembled on the modified CB layer? Will the tissue paper peel off after long-term testing?

Reply 2: We thank the reviewer for the comments. The tissue layer and the modified carbon black (CB) layer in the electricity generation unit are assembled by the surface tension of water and the capillary effect. First, both the tissue layer and the CB layer are wetted by

water. Then the tissue layer is controlled to connect the CB layer. Under the minimization of systematical surface energy, the tissue paper can assemble on the modified CB layer spontaneously and closely.

Due to the capillary wicking effect and the nearly saturated relative humidity inside the cell, the bilayer structure can continuously wet. Therefore, the tissue paper maintains the assembling state and would not peel off after the long-term test.

Manuscript modification:

Page 4, line 67, the description “by the surface tension of wet surfaces” is added to explain the assembling mechanism of the bilayer structure.

Comment 3: The hermetic cell without tissue sharply drops to 0 V in less than 1000 s after being sealed. The voltage of the hermetic cell with tissue flatly decreases from 1 V to 0.16 V and keeps stable after being sealed. Why does the voltage of the hermetic cell with tissue keep stable after being sealed? What’s the function of the tissue paper? Does the continuous evaporation remain in the hermetic cell without tissue?

Reply 3: We thank the reviewer for the comments. For the hydrovoltaic electricity, there are two kinds of mechanism, one is the streaming potential that relies on the evaporation-induced capillary flow, and the other is evaporation potential that relies on water content gradient and evaporation.

For the cell without tissue, before sealing, water will evaporate from the carbon layer with a large evaporation rate ($\sim 0.2 \text{ kg/m}^2\text{h}$) to induce streaming potential. Meanwhile, due to rapid evaporation, the water content gradient will form on the carbon layer to generate evaporation potential. Therefore, the cell without tissue can generate an open-circuit voltage of about 3 V.

When the cell without tissue is sealed, the evaporation rate from the carbon layer will greatly reduce ($\sim 10^{-3} \text{ kg/m}^2\text{h}$), and thus the streaming potential can be neglected ($\sim 1 \text{ mV}$). Meanwhile, due to the superhydrophilicity (Fig. R1) of the layer, the ultrahigh specific surface area of the porous carbon, and the high relative humidity inside the cell, a water layer would form on the carbon layer (Fig. R2a), which eliminates the water content gradient. Therefore, the hermetic cell without tissue sharply drops to 0 V after being sealed.

Fig. R1 The contact angles of the modified CB layer to show its superhydrophilicity.

For the cell with tissue, before sealing, water will evaporate from both the carbon layer and the tissue layer to generate streaming potential. Meanwhile, due to rapid evaporation, water content gradient will form on the carbon layer to generate evaporation potential. However, the existence of tissue layer reduces the evaporation rate from the carbon layer, which lowers the output performance of the cell (1.2 V verse 3 V).

When the cell with tissue is sealed, due to the water evaporation and gravity effect, water content gradient would form along the tissue. Besides adsorbing water from bulk through capillary wicking, the tissue can adsorb water from the adjacent carbon layer, thus induces water content gradient along the carbon layer (Fig. R2b). This causes the generation of evaporation potential, and accordingly stable output.

Fig. R2 The diagram to show (a) the water film on the CB layer, and (b) the water content gradient induced via water adsorption by the tissue layer.

The function of tissue paper:

1. Assisting evaporation in such a hermetic space with high relative humidity.
2. The tissue paper can adsorb water from the adjacent carbon layer, which induces a water content gradient inside the carbon layer to generate evaporation potential and output power.

Due to the ambient temperature fluctuations, continuous water evaporation remains in the hermetic cell without tissue (Fig. R3). However, the evaporation rate ($\sim 10^{-3}$ kg/m²h, compared to ~ 1 kg/m²h of an open cell) is small enough to generate obvious streaming potential. Meanwhile, water content gradient cannot from on the carbon layer to induce evaporation potential. Therefore, even there exists continuous evaporation, no power will be generated (Fig. R4).

Fig. R3 The evaporation measurement of the hermetic cell without tissue.

Manuscript modification:

1. The Fig. R2 is added to the Supporting Materials in Fig. S6.
2. Further discussion on the functions of tissue paper is added the Manuscript Page 13.

Comment 4: “a short-circuit current of 200 nA...in Fig. 1c.” The figure only shows the short-circuit current from 159 h to 160 h.

Reply 4: We thank the reviewer for the comment. The open-circuit voltage and the short-circuit current cannot be tested simultaneously, because of the mutually exclusive circuit design. Therefore, we measure the open-circuit voltage and the short-circuit current orderly. Firstly, long-term open-circuit voltage measurement is conducted to demonstrate the sustainability and stability of the HHC. Subsequently, short-circuit current measurement is performed to illustrate the output performance of the device. This is a common practice in previous reports¹⁻³. In our manuscript, we monitored the open-circuit voltage for 160 hours, then modified the circuit design and proceeded with a 1-hour short-circuit current measurement.

Comment 5: The water evaporation is induced by the ambient temperature fluctuation. When the ambient temperature is managed to be constant, will the water evaporation and hydrovoltaic phenomenon disappear?

Reply 5: We thank the reviewer for the constructive suggestions. Theoretically, when the ambient temperature is managed to be ideally constant, the evaporation and the hydrovoltaic effect would vanish. This can be explained that without ambient temperature fluctuation, the temperature distribution inside the HHC will be uniform, and no vapor concentration gradient will be induced. Under this situation, there is no net evaporation, as well as no electricity generation. To verify this inference, an additional experiment is designed and performed. The ice-water bath is used to serve as environment with nearly constant temperature (Fig. R4a and b). Initially, the open-circuit voltage of the HHC is 143 mV at ambient environment, and after placing to the ice-water bath, the voltage drops to 19 mV in 2 hours and remains stable (Fig. R4c).

The results can be explained as follow.

In the first hour after placing to the ice-water bath, the temperature of the container wall decreases to 0 °C, while the wet air inside the container remains warmer. Therefore, temperature gradient is formed between the bilayer regions and the wall. Water evaporates from the warmer electricity generation unit, and condensates on the colder walls of the container. Therefore, the voltage drops slowly in the first hour. When the temperature of the whole system approaches gradually to 0 °C and remains stable, the temperature gradient inside the container, and also the electricity, minimizes. The final open-circuit voltage of 19 mV may result from the ineludible fluctuations of ice melting.

Fig. R4 The experiment design (a) and (b), and the open-circuit voltage monitoring at nearly constant 0 °C.

Manuscript modification:

1. Fig. R4 is added to the Supporting Materials as Fig. S12.
2. The description of the experiment of Fig. R4 is added to the Manuscript Page 10.

Comment 6: The authors are suggested to perform composition analysis for the wicking bilayer before and after long-term energy harvesting to see whether any change in the chemical composition. I'm wondering whether there is some chemical reactions take place within the container.

Reply 6: We thank the reviewer for the comment. With supplement experimental results and discussions, it is concluded that no intrusive reaction takes place in the HHC. Firstly, the modified carbon materials^{1,4-8} and the cellulosic materials^{4,9-13} are the common materials in hydrovoltaic electricity generation. These materials have been ruled out the possibility of chemical reactions in the previous studies in principle. Also, we add the characterizations of X-ray photoelectron spectroscopy (XPS) on the CB layer and the tissue paper before and after the electricity generation process (Fig. R5). The results indicate that

the elementary compositions and the binding ways do not change in the electricity generation process.

Fig. R5 The XPS measurements of the bilayer structure. The XPS full spectrum analysis of (a) the tissue paper layer, and (b) the modified CB layer before and after the electricity generation process. (c) The binding energy in the PEI modification of CB layer.

Manuscript modification:

The Fig. R5 is added to the Supporting Materials as Fig. S4.

Comment 7: Since the ambient temperature fluctuation is critical to maintain the electricity harvesting, the authors need to carefully compare the electric signal (e.g., figure 1c) to the variation trend of ambient temperature (e.g., figure 2d). Is there any correlation between the two variables? Of not, what is the reason?

Reply 7: We thank the reviewer for the comments. In Supporting Materials 3.2 and 3.3, the theoretical and numerical derivations explain that two key factors contribute to the evaporation phenomenon inside the HHC: the changing rates of the ambient temperature, and the hysteric in heat conductivity of different parts of the hermetic device. During the

testing process, the overall ambient temperature changes in similar slow rates, so the HHC has a stable open-circuit voltage, as demonstrated in Fig. 1c and Fig. 3b in the manuscript.

For the correlation between the ambient temperature and the electric signals, we amplify the amplitudes of the variables, as shown in Fig. 3b and c in previous submission (also shown in Fig. R6). Under the temperature fluctuations, the HHC generates similar open-circuit voltage at different temperature, since the water gradients concentration in the bilayer are similar with similar ambient temperature changing rates (Fig. R6a). Although the open-circuit voltage remains unchanged at different ambient environments, the voltage displays an obvious fluctuation when the temperature increases sharply from 15 to 48 °C in 40 minutes (Fig. R6b). This can be explained that the drop-off of relative humidity accelerates the water evaporation and alters the water distribution in the electricity generation unit, thus enlarging the voltage. The return of the voltage shortly after the increase is attributed to that the RH restores equilibrium and the moisture content gradient restabilizes. More importantly, the result evidences that the open-circuit voltage can remain stable under ambient temperature fluctuations.

Fig. R6 (a) The open-circuit voltages and the short-circuit currents at different ambient temperatures, shown in Manuscript Fig. 3b. (b) The voltage increases rapidly and then restores with a dramatic change in temperature, shown in Manuscript Fig. 3c.

Comment 8: In Page 24, “The mass change...in Fig. 2d.” The data does not match figure.

Reply 8: We apologized for the reference error in Page 24. “The mass change...in Fig. 2d.” should refer to the Fig. 2c, which is corrected in the revised manuscript.

Manuscript modification:

Page 24: the sentence “The mass change...in Fig. 2d.” is replaced by “The mass change...in Fig. 2c.”

Comment 9: In Fig. 2g and h, why does the temperature gradient between bulk water and the bilayer decrease at high ambient temperature, while the temperature gradient between the container and the bilayer increases? Does the size of the container influence the performance of HHC?

Reply 9: We thank the reviewer for the comments. For the simulation results, the driven temperature difference during both the temperature rise ($T_{\text{bilayer}}-T_{\text{bulk_water}}$) and temperature drop ($T_{\text{bilayer}}-T_{\text{container}}$) decreases at high ambient temperature. The temperature gradient is caused by the low thermal conductivity of air and the large specific heat capacity of bulk water. The ambient temperature fluctuations show negligible influence on the specific heat capacity of bulk water. However, when ambient temperature rises, the vapor concentration (the saturated vapor pressure, p) in the air remarkably increases (Table R1), which enlarges the heat conductivity of air (λ_a), and thus reduces the temperature gradient inside the HHC. Similarly, when ambient temperature drops, the vapor concentration (the saturated vapor pressure, p) in the air decreases, which remarkably lowers the heat conductivity of air (λ_a), and thus promotes the temperature gradient inside the HHC. Therefore, at high ambient temperature, both the temperature gradient between bulk water and the bilayer ($T_{\text{bilayer}}-T_{\text{bulk_water}}$) and the temperature gradient between the container and the bilayer ($T_{\text{bilayer}}-T_{\text{container}}$) decreases.

Table R1. The values of the parameters change with the ambient temperature, where p is the saturated vapor pressure, λ_a is the heat conductivity of the air at different temperature.

$T/^\circ\text{C}$	10	20	30	40
p/Pa	1312	2339	4246	7381
$\lambda_a/(\text{mW} \cdot \text{m}^{-1} \cdot \text{K}^{-1})$	25.1	25.9	26.6	27.4

The size of the container would influence the performance of HHC. Based on the two dimensional model in Manuscript Page 8, $\dot{m} \sim \frac{DMpLC_{p,a}\rho_a x_w}{\lambda R^2 T^3} \frac{\partial T}{\partial t}$, where the evaporation rates

is proportional to the diameter of the container (detailed information in Supporting Materials 3.3). Further, additional three-dimensional simulations are performed to evaluate the influence of the container volume on the performance of the HHC. The cylindrical container is scaled in all dimensions at the ratio of 1.25, 1.5, 1.75, and 2, varying in volumes from the original size to 8 times larger, with the same electricity generation unit. Fig. R7 shows the positive correlations between the size of the container and the evaporation rates, which matches the results from the two-dimensional model. Also, the influence of the container size is more obvious when the ambient temperature increases than decreases. This can be explained as follow:

- For the larger containers, the area of the bulk water increases, while the area of the electricity generation unit remains unchanged.
- When the ambient temperature raises, the temperature gradient between the bilayer and the bulk water ($T_{\text{bilayer}} - T_{\text{bulk_water}}$) drives the evaporation from the electricity generation unit. The condensation on the surface on the bulk water increases with larger area, promoting the evaporation on the unit.
- When the ambient temperature drops, the temperature gradient between the bilayer and the wall of the container ($T_{\text{bilayer}} - T_{\text{container}}$) drives the evaporation on the unit. The evaporation on the warmer surface of the bulk water increases with larger area, which competes with the evaporation on the unit, therefore restrain the evaporation on the unit.

Hence, the impact of the container size on the evaporation differs while the temperature increasing and decreasing.

Fig. R7. The difference of the simulated evaporation rates in different sized containers when the ambient temperature (a) increasing and (b) decreasing.

Manuscript modification:

1. Fig. R7 is added to the Supporting Materials as Fig. S20.
2. The discussion on Fig. R7 is added to the Supporting Materials 3.2.

Reviewer #2 (Remarks to the Author):

The author proposed a hydrovoltaic effect inside the hermetic cell. However, the reviewer is confused about some points in operation mechanism and evaluation as an energy harvesting technology. The reviewer can only judge the manuscript after checking some important experimental results suggested below.

We thank the reviewer for the time and efforts in reading our manuscript. We believe that with more experimental results and necessary discussion provided in the revised manuscript, it is more likely for you to judge our work. In the following, we have addressed the concerns point by point.

Comment 1: At 100% relative humidity, although there may be a dynamic equilibrium of evaporation and condensation of water, this equilibrium theoretically exists at any point in the environment. It has nothing to do with the material structure. Why can the container exhibit evaporation in one area and condensation in one area?

Reply 1: We thank the reviewer for the comment. The reviewer is right that at an ideally 100% relative humidity, the dynamic equilibrium of evaporation and condensation theoretically exists at any point in the environment. However, in practice, due to fluctuations in ambient temperature (Fig. S10 in Supporting Materials) and the hysteric in heat conductivity of different parts of the hermetic device, there exists temperature gradient inside the hermetic container, and the inside relative humidity cannot stabilize at 100%^{14,15}. For illustration, we supplement the temperature distribution data by monitoring the temperature near the bilayer, the temperature in the bulk water, the temperature of the wall, as schemed in Fig. R8. The results provide strong evidence that ambient temperature fluctuations will indeed cause temperature gradient inside the hermetic container.

Fig. R8 The inner temperature gradient in a hermetic container. (a) Diagram that shows the measuring method. (b) The records of the temperatures in different regions that form the gradient, and the calculated driven temperature difference.

From another aspect, the saturated vapor pressure of water highly depends on temperature. As described by Clausius-Claperon equation $\frac{dP}{dT} = \frac{L}{Tdv}$, the saturated vapor pressure could remarkably change at different temperatures. For example, it enlarges by 81% when temperature increases from 20 °C to 30 °C. This leads to varying in water vapor content/concentration inside the container, generating vapor diffusion from high concentration (high temperature) area to low concentration (low temperature) area. During temperature rise process, the bilayer region is warmer than the bulk water (Fig. R9a), allowing the water vapor near the bilayer to diffuse towards the bulk water, facilitating evaporation from bilayer and condensation to bulk water. Similarly, during temperature drop process, the bilayer region is warmer than the container wall (Fig. R9b), allowing the water vapor near the bilayer to diffuse towards the container wall, facilitating evaporation from bilayer and condensation to the wall. Therefore, water can continuously evaporate from bilayer due to the temperature fluctuations.

Fig. R9 The simulated dependence of the evaporation rates on the driving temperature differences at varying ambient temperatures, shown in Manuscript Fig. 2g and h.

Meanwhile, supplement experiments are designed to confirm that such evaporation phenomenon does happen inside hermetic containers (Fig. R10, as Fig. 2c in our previous submission). The experimental design is illustrated in Fig. R11.

- Glass beakers are upturned with the bottom fully immersed in water to construct water-sealed hermetic environment.
- Inside the beaker, a sealed container with water at the bottom is placed on a superhydrophobic support to avoid contact between the container and the beaker.
- A tissue paper is lifted by a superhydrophobic holder, and one lower-end of the tissue is immersed in the water inside the container, through a slit on the top of the container that guarantee the mass loss of the container can be only caused by evaporation from the tissue paper.
- Six sets of such units are prepared (Fig. R11b). The container with tissue paper and superhydrophobic holder (Fig. R11c) is taken from the beaker and weighted with a time-interval of 12 hours.

The mass losses of the HHC are summarized in Fig. R12. We can see that the mass loss of the container enlarges linearly with evaporation time. Therefore, it is concluded that water can continuously evaporate from the tissue paper inside our hermetic design. Note that the linear trend can exclude the influence from the humidity and evaporation unsaturation during the water-sealing process.

Fig. R10 The mass loss on the tissue paper in the hermetic container, mentioned in the Manuscript as Fig. 2c.

Fig. R11 The diagrams of the experiment design. (a) The structure of the water-sealed evaporator. (b) Six sets are tested sequentially. (c) The detailed structure of the evaporator.

In addition, another experiment is performed to prove that the evaporation happens because of the ambient temperature fluctuations. We employ the ice-water bath method to create an environment with nearly constant temperature for the HHC. Initially, the open-circuit voltage of the HHC is 143 mV at ambient environment, and after placing to the ice-water bath, the voltage drops to 19 mV in 2 hours and remains stable (Fig. R12).

The results can be explained as follow. In the first hour after placing to the ice-water bath, the temperature of the container wall decreases to 0 °C, while the wet air inside the container remains warmer. Therefore, temperature gradient is formed between the bilayer

regions and the wall. Water evaporates from the warmer electricity generation unit, and condensates on the colder walls of the container. Therefore, the voltage drops slowly in the first hour. When the temperature of the whole system approaches gradually to 0 °C and remains stable, the temperature gradient inside the container, and also the electricity, minimizes. The final open-circuit voltage of 19 mV may result from the ineludible fluctuation of ice melting.

Fig. R12 (a) and (b)The experiment design, and (c) the open-circuit voltage monitoring at nearly constant 0 °C.

Supported by these experimental results and discussion, it is concluded that the bilayer can indeed evaporate continuously inside such a hermetic container, which results from the ambient temperature fluctuations.

Manuscript modification:

1. Fig. R8b replaces Fig. 2d. Fig. R8a is added to the Supporting Materials Fig. S10.
2. The discussions on the additional experiments are added to the Manuscript Page 7 and 10.
3. The former Fig. 2d, Fig. R11 and Fig. R12 are added to the Supporting Materials as Fig. S8, Fig. S10 and Fig. S12.

Comment 2: What is the role of tissue, and how does it play the role?

Reply 2: We thank the reviewer for the comments.

There are two roles of the tissue paper in the HHC device:

1. The tissue paper can achieve evaporation in such a hermetic space with high relative humidity.
2. The tissue paper can adsorb water from the adjacent carbon layer, which induces a water content gradient inside the carbon layer to generate evaporation potential and output power.

The way that the tissue paper plays its role can be explained from the mechanism behind the electricity generation by the HHC.

Generally, there are two kinds of mechanism account for the hydrovoltaic electricity, one is the streaming potential that relies on the evaporation-induced capillary flow, and the other is evaporation potential that relies on water content gradient and evaporation.

If the cell does not have tissue, before sealing, water will evaporate from the carbon layer with a large evaporation rate ($\sim 1 \text{ kg/m}^2\text{h}$) to induce streaming potential. Meanwhile, due to rapid evaporation, the water content gradient will form on the carbon layer to generate evaporation potential. Therefore, the cell without tissue can generate an open-circuit voltage of about 3 V (Fig. R13).

Fig. R13 The open-circuit voltage of the HHC changes with and without the tissue paper after sealing, mentioned in the Manuscript Fig. 3i.

When this kind of cell is sealed (the hermetic cell without tissue), the evaporation rate from the carbon layer will greatly reduce ($\sim 10^{-3} \text{ kg/m}^2\text{h}$), and thus the streaming potential can be neglected ($\sim 1 \text{ mV}$). Meanwhile, due to the superhydrophilicity of the layer (Fig.

R14, as Fig. S4 in the Manuscript), the ultrahigh specific surface area of the porous carbon (SEM), and the high relative humidity inside the cell, a water layer would form on the carbon layer, which eliminates the water content gradient. Therefore, the hermetic cell without tissue cannot continuously generate electricity (Fig. R13).

If the tissue is assembled into the cell, before sealing, water will evaporate from both the carbon layer and the tissue layer to generate streaming potential. Meanwhile, due to rapid evaporation, water content gradient will form on the carbon layer to generate evaporation potential. However, the existence of tissue layer reduces the evaporation rate from the carbon layer, which lowers the output performance of the cell (1.2 V verse 3 V).

When this kind of cell is sealed (the hermetic cell), due to the water evaporation and gravity effect, water content gradient would form along the tissue. Besides adsorbing water from bulk through capillary wicking, the tissue can adsorb water from the adjacent carbon layer, thus induces water content gradient along the carbon layer. This causes the generation of evaporation potential, and accordingly stable output.

Therefore, the tissue plays a significant role for the continuous electricity generation by the HHC.

Fig. R14 The contact angles of the modified CB layer, shown in Manuscript Fig. S4.

Manuscript modification:

The discussion on the roles of the tissue paper is added to the Manuscript Page 13.

Comment 3: Whether can the authors provide the relevant experimental data of temperature gradient? What is the causal relationship between temperature gradient and evaporation condensation?

Reply 3: Thank the reviewer for the comments. We have performed the direct temperature monitoring for 24 hours at locations including near the bilayer, in the bulk water, on the

container wall, and the ambient environment, as schemed in Fig. R8 of Reply1. The results in Fig. R8b clearly shows that there are fluctuations in ambient temperature, and also indeed exists temperature gradients inside the hermetic container.

The water evaporation from the bilayer results from temperature gradient inside the hermetic cell. The net evaporation rate positively depends on the temperature gradient, as explained in detail by the simulation of water circulation and the mechanism of water evaporation inside the hermetic HHC (Supporting Materials 3.2 and 3.3). And faster water evaporation generates larger electricity output. This conclusion can be evidenced by the following experimental results. We amplify the amplitudes of the ambient temperature gradient by increasing temperature from 15 to 48 °C in 40 minutes, and monitor the open-circuit voltage (Fig. R15). We can see that the voltage displays an obvious fluctuation during the warming process, and then returns to 150 mV when the temperature stabilizes at 48 °C. Compared to the change of ambient temperature, the recovery of voltage shows a hysteresis of about 60 minutes, which results from the reconstruction of equilibrium in relative humidity and restabilization in moisture content gradient inside the hermetic container.

For the fluctuations in ambient temperature, as the caused temperature gradient is small enough and there exists hysteresis of the heat conductivity in the closed system, the electricity generation performance of the HHC can maintain stable, as shown in Fig. R15.

Fig. R15 (a) The open-circuit voltages and the short-circuit currents at different ambient temperatures, shown in Manuscript Fig. 3b. (b) The voltage increases rapidly and then restores with a dramatic change in temperature, shown in Manuscript Fig. 3c.

Manuscript modifications:

The Fig. R8b, as mentioned in Reply 1, is added to the Manuscript as Fig. 2d.

Comment 4: Is it possible that the experimental results in FIG. 2b and c were caused by the fact that tissue and water did not reach the equilibrium of water absorption in the initial stage? So the tissue continues to absorb water resulting in a decrease in the solution? Why doesn't the condensed water return to the original container?

Reply 4: We thank the reviewer for the comments. The experimental design in Fig. 2b and c is illustrated in Fig.R11, mentioned in Page13, Reply 1.

- Glass beakers are upturned with the bottom fully immersed in water to construct water-sealed hermetic environment.
- Inside the beaker, a sealed container with water at the bottom is placed on a superhydrophobic support to avoid contact between the container and the beaker.
- A tissue paper is lifted by a superhydrophobic holder, and one lower-end of the tissue is immersed in the water inside the container, through a slit on the top of the container that guarantee the mass loss of the container is only caused by evaporation from the tissue paper.
- Six sets of such units are prepared (Fig. R11). The containers with tissue paper and superhydrophobic holder are subsequently taken from the beaker and weighted with a time-interval of 12 hours. The mass losses of the containers are summarized in Fig. R10.

The states that “the tissue and water did not reach the equilibrium of water absorption in the initial stage” is right, while this process will only last for a few hours. This can be explained that the for the 72-hour testing of the 6 beaker units, the mass loss linearly increase with evaporation time. If the mass loss results from the non-equilibrium of water absorption, the loss rate will reduce over time and not keeps constant. Meanwhile, the container is sealed with a slit for the tissue stick out, to guarantee that water in the container can only reduce through wicking by the tissue and evaporating in the beaker. And also, before the container is weighted, the water condensed on the container surface is also removed. Therefore, it is concluded that the mass loss in Fig. 2b and c originates from the water evaporation in the hermetic beaker.

In our experiment design, to avoid condensed water return to the original container, we use a sealed container. A slit is prepared on the top of the sealed container to allow the tissue stick out and water can wick along the tissue for evaporation.

Manuscript modifications:

The Fig. R11, as mentioned in Reply 1, is added to the Supporting Materials as Fig. R12.

Comment 5: If the energy output of the device comes from the heat energy in the environment, when the entire system is placed in an adiabatic environment, and the electrical output is outside the adiabatic environment, will the temperature inside the device reduced like a refrigerator?

Reply 5: We thank the reviewer for the comment. The temperature inside the HHC would not decrease when placed in an adiabatic system. The energy output of the HHC comes from the heat energy in the environment, achieved by ambient temperature fluctuations. If the entire system is placed in an adiabatic environment, the temperature fluctuations will disappear, and also there will be no temperature gradient inside the hermetic HHC. That is, the whole device would reach absolute equilibrium in temperature and evaporation-condensation. As a result, no electricity will be generated by the device, and the temperature inside the HHC will not reduce.

Manuscript modification:

The discussion on energy conversion efficiency is added to the Supporting Materials 3.5.

Reviewer #3 (Remarks to the Author):

The authors present a truly unique system of hydrovoltaic that works under conditions that other hydrovoltaic do not; hermetically sealed. This type of system could have a great number of advantages in actually deployment of devices since other technologies would be strongly affected by environmental conditions while these would not.

We thank the reviewer for the time and efforts in reading our manuscript and raising positive evaluation of “a truly unique system”, “have a great number of advantages in actually deployment”. In the following, we have addressed the concerns from the reviewer point by point.

Comment 1: The methodology and work are sound, however, I think there is a major opportunity missed herein. One question that remains in hydrovoltaic is a measure of the efficiency of thermal energy conversion. With this sealed system the temperature changes induced by evaporation and the resulting energy conversion could possibly be used (almost like a bomb calorimeter) to measure the quantum efficiency of the energy conversion process. I would expect no less from an article in Nature, to be even more impactful.

Reply 1: We thank the reviewer for the comments and suggestion. The hermetic hydrovoltaic cell (HHC), serving as a closed system, requires continuous energy input from the ambient temperature fluctuation to construct the inside temperature gradient and further the evaporation. In an adiabatic system (like a bomb calorimeter), the HCC is supposed to reach an absolute balance of thermodynamic with temperature gradient and evaporation, As a result, the HHC will stop work and the energy conversion will not be realized.

In addition, a supplement experiment is designed to imitate a HHC surrounding with a constant ambient temperature (Fig. R16), by covering the HHC with ice-water bath. Initially, the open-circuit voltage of the HHC is 143 mV at ambient environment, and after placing to the ice-water bath, the voltage drops to 19 mV in 2 hours and remains stable (Fig. R16c). The results can be explained as follow. In the first hour after placing to the ice-water bath, the temperature of the container wall decreases to 0 °C, while the wet air inside the container remains warmer. Therefore, temperature gradient is formed between the bilayer regions and the wall. Water evaporates from the warmer electricity generation

unit, and condensates on the colder walls of the container. Therefore, the voltage drops slowly in the first hour. When the temperature of the whole system approaches gradually to 0 °C and remains stable, the temperature gradient inside the container, and also the electricity, minimizes. The final open-circuit voltage of 19 mV may result from the ineludible fluctuations of ice melting.

Fig. R16 (a) and (b) The experiment design. (c) The open-circuit voltage monitoring at nearly constant 0 °C.

Manuscript modifications:

1. A detailed discussion of energy conversion efficiency is added to Supporting Materials 3.5.
2. The Fig. R16 is added to Supporting Materials as Fig. S12.

Comment 2: The authors did not compare their devices to other devices that have been reported and have orders of magnitude higher power outputs. This report is scientifically interesting but it is a bit misleading to report this device as comparable with high performance hydrovoltaics. With respect to the above discussion on efficiency of this device, I would like to see a lengthy discussion of how these sealed devices cannot provide comparable power to devices that are open to the environment. Maybe long term one could

make that argument but these are really not powerful, especially considering the size of the device.

Reply 2: We thank the reviewer for the comments. We do notice that several hydrovoltaic devices hold higher power output¹⁶⁻¹⁸ than the open devices referenced in Manuscript Fig. 1d. Since these devices employ composite strategy by combining the hydrovoltaic effect and the galvanic effect (by using reactive metal as electrodes), we do not consider these devices fair for comparison. The impacts of the electrodes are discussed in Manuscript Fig. 3f, where our HHC strategy could generate multiple power by right of the galvanic effect. More details of short-circuit current are provided in Fig. R17. Therefore, these devices with higher output are not listed in Fig. 1d.

Fig. R17 The output of the HHC with electrodes of different materials.

A discussion on the energy conversion efficiency is appended in the Supporting Materials to further explain the correlation between the output of the devices and the evaporation rates, including the summary of the reported hydrovoltaic generators. The discussion is appended below.

The efficiency of the HHC and other hydrovoltaic generators

The electricity generation of hydrovoltaic devices depends on the water evaporation induced by ambient heat. Unlike other thermal engines that output power from high grade energy sources like petroleum and liquefied gas, it's difficult to calculate the input energy (including the ambient heat and the solar energy) for the hydrovoltaic devices, as well as the energy efficiency $\eta = W/Q_{input}$, where η is the efficiency, W is the output, and the

Q_{input} is the overall input energy. Therefore, recent research progresses generally uses the open-circuit voltage and/or short-circuit current as a measurement of the output performance.

Meanwhile, from the perspective of energy conversion process, the input energy is the low-grade ambient heat energy, which converts into the internal energy of vapor, the electricity for output, and the wasted energy. Therefore, theoretically, the efficiency of hydrovoltaic generators should be calculated by $\eta = \frac{W_e}{W_e + E_w + W_d}$, where W_e is the electricity output, E_w is the internal energy of vapor, and W_d is the dissipation energy such as viscous resistance of water that can be neglected. Therefore, the efficiency of hydrovoltaic generators can be simplified as

$$\eta \approx \frac{W_e}{W_e + E_w} = \frac{VIt}{VIt + 2L\dot{m}tA} = \frac{VI}{VI + 2L\dot{m}tA}$$

where I is the short-circuit current, V is the open-circuit voltage, and t is the time, L is the latent heat of water, \dot{m} is the evaporation rates, A is the area of the devices. For the hydrovoltaic generators, I lies in the scale of μA , V lies in the scale of V , L is 44.2 kJ/mol , \dot{m} lies in the scale of $\text{kg/m}^2\text{h}$, A lies in the scale of 10 cm^2 . Therefore, the electricity (with the power of μW) is far from comparable with the internal energy of vapor (with the power of W), generating a very small efficiency.

Many representative hydrovoltaic devices are listed below with their performances, including open-circuit voltages, short-circuit currents, evaporation rates, and the combining methods. Note that several hydrovoltaic-related reports do not provide water evaporation data. For these devices, as no special structures are designed to promote evaporation, the evaporation rate is regarded as $0.22 \text{ kg/m}^2\text{h}$, the evaporation rate on clam water surface. With the reported data, we could calculate the approximate energy conversion efficiency for some devices, shown in Table S4.

Table R2. The reported hydrovoltaic devices and their performances.

Voltage (V)	Current (μA)	Evaporation rate ($\text{kg}/(\text{m}^2\text{h})$)	Combining methods	Efficiency ($10^{-4}\%$)	Reference
----------------	------------------------------	--	----------------------	-------------------------------	-----------

0.4	14	-	-	15.6	19
0.28	55	-	-	-	20
1.2	0.49	-	-	1.96	21
0.3	100	-	Galvanic effect	-	22
0.084	500	1.1	Galvanic effect	2.68	23
0.25	-	0.2	-	0.102	24
0.24	42.7	-	Moisture adsorption	18.9	25
0.74	-	-	Galvanic effect	1.66	26
0.16	20	-	Galvanic effect	-	27
2.5	0.4	-	-	0.83	28
1	0.1	-	-	1.11	1
0.432	64.2	2.78	-	3.46	29
0.5	0.25	-	-	-	3
1.48	0.037	-	Ethanol evaporation	-	30
0.71	0.38	1.866	Galvanic effect	0.30	31
0.6	0.12	-	-	0.20	31
0.37	-	1.3	-	10.1	32
0.55	22	1.15	-	-	12
0.31	5.3	1.3	-	0.31	33
0.37	-	1.3	-	-	32
0.36	13	2.78	-	0.68	34
0.778	32	1.74	Galvanic effect		35
0.022	144	2.26	-	0.16	36
0.73	0.6	2.38	-	0.045	37
0.16	0.2	10 ⁻³	-	31.3	Our work

Though the open-circuit voltages and the short-circuit currents of the devices differ by more than hundredfold, regardless of the devices with combining methods, taking the size of the devices into consideration, most devices in Table S4 have similar energy conversion efficiencies. The results can be explained from the aspect of energy generation origin. For these devices, the electricity comes from the water molecular interacting with the charged surface, sharing the same mechanism. Therefore, these devices generate electricity from evaporation process with the energy conversion efficiency of a similar order of magnitude. Limited by its hermetic structure, unlike other devices in Table S4, the HHC holds an evaporation rate of $\sim 10^{-3}$ kg/(m²h), which is 1000 times smaller than the others. This lowers hydrovoltaic effect, including the streaming potential that relies on the evaporation-induced capillary flow, and the evaporation potential that relies on water content gradient and evaporation. Therefore, with the similar efficiencies, our HHC method inevitably generates lower outputs. Even though, the HHC performs better in other important aspects, including the range of applications and the water consumption.

Manuscript modifications:

The discussion on the energy conversion efficiency and Table R2 are added to the Supporting Materials 3.5.

Comment 3: There are a few minor issues with spelling and grammar where the manuscript might benefit from another round of editing. Also, I am not sure if it a sign convention difference or if the authors are using absolute values for current and voltage but these would typically be different signs. Could the authors explain this?

Reply 3: We thank the reviewer for pointing out these minor issues, and also the comments about the electric signs. We have corrected these issues, and doublechecked the manuscript to avoid typos. The open-circuit voltage and the short-circuit current are abbreviated as “Voltage” and “Current” in the figures, and “ V ” and “ I ” in equations. Meanwhile, to make our manuscript more readable, we try to keep our data in a positive value, using absolute values for current and voltage. In our manuscript, we choose the bottom electrode as the positive electrode, and accordingly measure the results and reveal the mechanism.

Manuscript modifications:

The spelling mistakes are corrected in the Manuscripts.

Reviewer #4 (Remarks to the Author):

We thank the reviewer for the time and efforts in reading our manuscript. Hope our revised manuscript satisfies your scruple on this work.

References:

- 1 Xue, G. *et al.* Water-evaporation-induced electricity with nanostructured carbon materials. *Nat. Nanotechnol.* **12**, 317-321 (2017).
- 2 Liu, X. *et al.* Microbial biofilms for electricity generation from water evaporation and power to wearables. *Nat. Commun.* **13**, 4369 (2022).
- 3 Liu, X. *et al.* Power generation from ambient humidity using protein nanowires. *Nature* **578**, 550-554 (2020).
- 4 Li, J. *et al.* Surface functional modification boosts the output of an evaporation-driven water flow nanogenerator. *Nano Energy* **58**, 797-802 (2019).
- 5 Li, S. M. *et al.* Synergistic Effects of TiO₂ and Carbon Black for Water Evaporation-Induced Electricity Generation. *ACS Appl Mater Interfaces* (2024).
- 6 Zhao, F., Cheng, H., Zhang, Z., Jiang, L. & Qu, L. Direct Power Generation from a Graphene Oxide Film under Moisture. *Adv. Mater.* **27**, 4351-4357 (2015).
- 7 Liu, K. *et al.* Induced Potential in Porous Carbon Films through Water Vapor Absorption. *Angew Chem Int Ed Engl* **55**, 8003-8007 (2016).
- 8 Tian, B. K. *et al.* Integrating reduced graphene oxides and PPy nanoparticles for enhanced electricity from water evaporation. *Int J Smart Nano Mat*, 1-13 (2023).
- 9 Peng, P., Yang, F., Wang, Z. & Wei, D. Integratable Paper-Based Iontronic Power Source for All-In-One Disposable Electronics. *Adv Energy Mater* (2023).
- 10 Dao, V. D., Vu, N. H. & Choi, H. S. All day *Limnobium laevigatum* inspired nanogenerator self-driven via water evaporation. *J Power Sources* **448** (2020).
- 11 Li, X., Zhang, K., Nilghaz, A., Chen, G. X. & Tian, J. F. A green and sustainable water evaporation-induced electricity generator with woody biochar. *Nano Energy* **112** (2023).
- 12 Xiao, P. *et al.* Exploring interface confined water flow and evaporation enables solar-thermal-electro integration towards clean water and electricity harvest via asymmetric functionalization strategy. *Nano Energy* **68** (2020).
- 13 Beaton, G. C., Kumar, R., Neokleous, N., Liu, G. J. & Stamplecoskie, K. Gold nanoparticle decorated filter papers as hydrovoltaic devices. *Sustainable Energy & Fuels* **6**, 4645-4651 (2022).

- 14 Hoffmeyer, P., Engelund, E. T. & Thygesen, L. G. Equilibrium moisture content (EMC) in Norway spruce during the first and second desorptions. *Holzforschung* **65**, 875-882 (2011).
- 15 Thybring, E. E., Kymäläinen, M. & Rautkari, L. Experimental techniques for characterising water in wood covering the range from dry to fully water-saturated. *Wood Sci. Technol.* **52**, 297-329 (2017).
- 16 Huang, F. J. *et al.* A hybrid nanogenerator for collecting both water wave and steam evaporation energy. *Nano Energy* **110** (2023).
- 17 He, N. *et al.* Ion engines in hydrogels boosting hydrovoltaic electricity generation. *Energy Environ. Sci.* **16**, 2494-2504 (2023).
- 18 Yan, H. & Qi, R. Effect of Non-Noble-Metal Electrode on Moisture-Enabled Electric Generator. *Acs Appl Electron Ma* **5**, 5809-5813 (2023).
- 19 Wang, Z. *et al.* Unipolar solution flow in calcium-organic frameworks for seawater-evaporation-induced electricity generation. *J. Am. Chem. Soc.* **146**, 1690-1700 (2024).
- 20 Qin, Y. *et al.* Constant electricity generation in nanostructured silicon by evaporation-driven water flow. *Angew. Chem. Int. Ed. Engl.* **59**, 10619-10625 (2020).
- 21 Ma, Q. *et al.* Rational design of MOF-based hybrid nanomaterials for directly harvesting electric energy from water evaporation. *Adv. Mater.* **32**, e2003720 (2020).
- 22 Youm, J. *et al.* Highly increased hydrovoltaic power generation via surfactant optimization of carbon black solution for cellulose microfiber cylindrical generator. *Surf. Interfaces* **38**, 102853 (2023).
- 23 Yang, P. H. *et al.* Solar-driven simultaneous steam production and electricity generation from salinity. *Energy Environ. Sci.* **10**, 1923-1927 (2017).
- 24 Fang, S., Lu, H., Chu, W. & Guo, W. Mechanism of water-evaporation-induced electricity beyond streaming potential. *Nano Res. Energy* **3**, e9120108 (2024).
- 25 Li, P. *et al.* Multistage coupling water-enabled electric generator with customizable energy output. *Nat. Commun.* **14**, 5702 (2023).
- 26 Yun, T. G. *et al.* Ion-permselective conducting polymer-based electrokinetic generators with maximized utility of green water. *Nano Energy* **94**, 106946 (2022).
- 27 Li, L. H. *et al.* A novel, flexible dual-mode power generator adapted for wide dynamic range of the aqueous salinity. *Nano Energy* **85**, 105970 (2021).

- 28 Shao, C. *et al.* Large-scale production of flexible, high-voltage hydroelectric films based on solid oxides. *ACS Appl. Mater. Interfaces* **11**, 30927-30935 (2019).
- 29 Sun, Z. *et al.* Achieving efficient power generation by designing bioinspired and multi-layered interfacial evaporator. *Nat. Commun.* **13**, 5077 (2022).
- 30 Fang, S. M., Li, J. D., Xu, Y., Shen, C. & Guo, W. L. Evaporating potential. *Joule* **6**, 690-701 (2022).
- 31 Li, Z. *et al.* Polyaniline-Coated MOFs Nanorod Arrays for Efficient Evaporation-Driven Electricity Generation and Solar Steam Desalination. *Adv. Sci.* **8**, 2004552 (2021).
- 32 Hou, B. *et al.* Flexible and portable graphene on carbon cloth as a power generator for electricity generation. *Carbon* **140**, 488-493 (2018).
- 33 Hou, B. *et al.* Flexible graphene oxide/mixed cellulose ester films for electricity generation and solar desalination. *Appl. Therm. Eng.* **163**, 114322 (2019).
- 34 Wan, Y. *et al.* Bird's nest-shaped Sb₂WO₆/D-Fru composite for multi-stage evaporator and tandem solar light-heat-electricity generators. *Small* **20**, e2302943 (2024).
- 35 Ma, J. *et al.* Achieving solar-thermal-electro integration evaporator nine-grid array with asymmetric strategy for simultaneous harvesting clean water and electricity. *Adv. Sci.* **10**, e2303815 (2023).
- 36 Li, Z., Chen, D., Gao, H., Xie, H. & Yu, W. Reduced graphene oxide composite nanowood for solar-driven interfacial evaporation and electricity generation. *Appl. Therm. Eng.* **223**, 119985 (2023).
- 37 Ge, C. *et al.* Fibrous solar evaporator with tunable water flow for efficient, self-operating, and sustainable hydroelectricity generation. *Adv. Funct. Mater.* **34**, 2403608 (2024).

Reviewer #1 (Remarks to the Author):

The authors have addressed most of the issues I raised in the first round of review. However, several issues remain to be addressed before the manuscript can be accepted for publication.

We thank the reviewer for the efforts on reviewing our manuscript. Based on the comments, we have added additional experiments and more detailed discussions, as well as double-checked the manuscript. Benefiting from these constructive suggestions, we believe that the quality of this manuscript has been greatly improved. Sincerely thanks for your great help.

Comment 1: The measurement of short-circuit current from 159 h to 160 h may mislead the readers that the short-circuit current can last for 160 h. It should mention that the short-circuit current was measured immediately after the open-circuit voltage and lasted only for 1 hour.

Reply 1: We thank the reviewer for the comment. We agree that the details of the measurement in Manuscript Fig. 1c should be mentioned. Based on your suggestion, we have added the description “Note that the inset short-circuit current is measured after the long-term testing of voltage” in page 4, lines 74-76. Meanwhile, we perform the long-term testing of the short-circuit current using a new HHC device (Fig. R1). The result is added to Supplementary Information Fig. S2.

Fig. R1 The long-term short-circuit current performance of the HHC.

Manuscript revision:

Page 4, lines 74-76, and Supplementary Information S2, added the sentence “Note that the inset short-circuit current is measured after the long-term testing of voltage, and a long-term testing of current is provided in Fig. S2”.

Comment 2: For the comment 9, according to the author’s reply, the temperature gradient inside the HHC is determined by the change in heat conductivity of air induced by ambient temperature. Why does the temperature gradient between bulk water and the bilayer ($T_{\text{bilayer}} - T_{\text{bulk water}}$) increase and then decrease as the temperature arises?

Reply 2: We thank the reviewer for the comment. We regret that the variation tendency of the temperature gradient between bulk water and the bilayer ($T_{\text{bilayer}} - T_{\text{bulk water}}$) was not described clearly enough in our manuscript.

As explained in the previous Response, Fig. 2g and h in Manuscript are the results of the numerical simulation to clarify the temperature gradients in hermetic cells induced by ambient temperature fluctuations. To avoid presupposing the existing of temperature gradients, we do not set the inner temperature gradient as an initial condition, but set a uniform temperature in the whole system at the beginning of the simulation. Therefore, the temperature gradient ($T_{\text{bilayer}} - T_{\text{bulk water}}$) at Time 0 is 0.

In the first stage, the temperature gradient is gradually formed and enlarged with the increasing of the ambient temperature. Specifically, as the ambient temperature increases by 1.2 K/hour, the bulk water initially has a slower temperature increase than the bilayer. This is because the heat capacity of bulk water is larger than that of the bilayer. Therefore, temperature gradients enlarge as the temperature arises.

In the second stage, the decreasing of temperature gradient can be explained that the heat transfer approaches dynamic equilibriums during the ambient temperature rising. Note that the heat conduction equilibriums at different ambient temperature are different, since in Fourier’s law $q = -\lambda \frac{dT}{dx}$, the heat conductivity of wet air (λ) changes with temperature (Table R1, and details in Supplementary Information 3.3). With more water molecules evaporated into the air, the heat conductivity of the nearly saturated wet air raises, so the heat transfer between the environment and the inner parts faster. Therefore, the temperature gradients decrease with higher ambient temperature.

In the Manuscript Fig. 2, we plot the whole simulating results, including the forming of the temperature gradient at the beginning, to show how the temperature gradient is induced by the ambient temperature fluctuations. To avoid concerns, we have added the description “Here the system has a uniform temperature distribution in the beginning, and then temperature gradients emerge with changing ambient temperature.” in the figure caption of Fig. 2.

Table R1. The values of the parameters change with the ambient temperature, where p is the saturated vapor pressure, λ_a is the heat conductivity of the air at different temperature.

$T/^\circ\text{C}$	10	20	30	40
p/Pa	1312	2339	4246	7381
$\lambda_a/(\text{mW} \cdot \text{m}^{-1} \cdot \text{K}^{-1})$	25.1	25.9	26.6	27.4

Manuscript revision:

Page 29, lines 552-553, added the description “Here the system has a uniform temperature distribution in the beginning, and then temperature gradients emerge with changing ambient temperature”.

Comment 3: The authors are suggested to fix the format errors, such as Page 21, line 433.

Reply 3: We apologize for the format errors. The Manuscript is rechecked to avoid typos.

Manuscript Modifications:

The format error has been corrected.

Reviewer #2 (Remarks to the Author):

The authors have provided a clearer explanation of the operation mechanism of the hermetic hydrovoltaic cell (HHC) by adding extensive experimental results and related analysis. Although the output performance of HHC is not considered outstanding compared to previously reported works, it does show unique advantages in the range of applications. However, the following comments need to be addressed before the manuscript could be recommended for publication:

We deeply thank the reviewer for the time and effort in thoroughly reading our manuscript and response. The quality of the manuscript has been greatly improved benefiting from your constructive comments in the *Round 1*. Here we have added additional experiments and detailed discussions. We believe that these revisions can fully address the issues from the reviewer.

Comment 1: It is recommended that the authors perform a long term test on the short circuit current of HHC, this data is important to assess the long term output stability of HHC.

Reply 1: We thank the reviewer for the suggestion. An additional long-term test on the short-circuit current of the HHC is added to the Supplementary Information to prove the long-term output stability of HHC (Fig. R2).

Fig. R2 The long-term short-circuit current performance of the HHC.

Manuscript revision:

The long-term short-circuit current performance of the HHC is added in Supplementary Information as Fig. S2.

Comment 2: As shown in Fig.4c, although the electrodes on the quartz plate of HHC are made by printing CB, it seems that they are connected out from the inside of the container to the outside with copper wires, does this part have an impact on the output performance?

Reply 2: We thank the reviewer for the comment. Our methods for connecting the devices will cause little impact on the output performance.

The possible reactions on electrodes come from the contacts between water and active electrodes. Generally, we connect the printed carbon electrodes into the circuits by both copper wires and copper clips, and adopt the hot melt glue to seal the copper from the possible condensed water (Fig. R3 a and b), which completely avoid the reactions. No copper component is immersed in or exposed to water. These two ways of connection show no difference on the output performance of the HHC (Fig. R3 c and d).

Fig. R3 a and b, The copper wire and clip are sealed by hot melt glue to avoid reactions with water. **c and d,** The selection of wire or clip does not affect the electricity performance.

Comment 3: In the bilayer structure, can an electrical signal also be generated continuously if the tissue does not completely cover the CB layer? For example, if the

tissue only covers the CB layer in the non-bulk water, what is the output performance of the HHC?

Reply 3: We thank the reviewer for the constructive comments. As described in our manuscript (Manuscript page 6, lines 115-120), the CB layer and the tissue layer both contribute to the output. Based on the comment, we design an additional experiment, in which the tissue paper is placed slightly above the bulk water (Fig. R4a). It is concluded that if the tissue paper does not fully cover the CB layer, there indeed exists continuously electrical signal in the HHC, but the voltage is lower than that of our HHC (Fig. R4b). The Fig. R4 is added to Supplementary Information as Fig. S9.

Fig. R4 The HHCs with different design and the open-circuit voltage of the devices.

For explanation, the adjustment of the tissue paper's position mainly affects the output performance by rebuilding the water content gradients in the CB layer and the tissue paper, like the experiment design in Manuscript Fig. 3h. As shown in Fig. R4, the tissue paper can create 84 mV, while the CB layer cannot generate electricity on itself due to the lack of water content gradient (Manuscript Fig. 3i). Here the bilayer structure is designed to enlarge the water content gradient and reach better output performance.

In this issue, the tissue paper is placed slightly above the bulk water, which cannot direct capillary wick from the bulk water. This design causes several impacts on the water content gradients in the bilayer. Firstly, the gradient in the CB layer enlarges. This is because that water content at the bottom of CB layer remains unchanged, while the water content at the top of CB layer decreases due to the larger adsorption amount by tissue layer. This causes a larger water content gradient in the CB layer, which generates a higher voltage by the CB layer. Secondly, the water content gradient in the tissue layer dramatically drops, since no capillary water directly comes from the bulk water. The decrease of the output in the tissue layer is more significantly than the increase in the CB layer. Combining both effects, the HHC with tissue paper placed slightly above the bulk water generates the open-circuit voltage of 98 mV, which is lower than that of the HHC in our manuscript.

Manuscript revision:

Fig. R4 is added to Supplementary Information as Fig. S9.

Comment 4: How does the pore size and thickness of the tissue affect the output performance of HHC?

Reply 4: We thank the reviewer for the comment. The structure of the tissue paper plays an important role in the capillary and the evaporation processes, which accordingly affect the output performance. In our manuscript, we use commercial tissues (Xuanzhi) to construct the bilayer. For comparison, we employed several kinds of commercially available tissue paper to fabricate HHC, which are the tissue paper of different brands, including Tempo, Grazie, and Elllair (noted as their brands for short), shown in Fig. R5.

Fig. R5 The open-circuit voltage of the HHCs with different brands of paper.

As illustrated in Fig. R5, all tissue papers can sustain the electricity generation, while the outputs differ with the paper brands. The paper that we employed in the Manuscript, Xuanzhi, a kind of Chinese traditional writing paper, achieves the largest output. The other kinds of tissue create lower voltage, 122 mV for Tempo, 64 mV for Grazie, and 52 mV for Elleair. To study the effects of the tissue structure on the outputs, by scanning electronic microscopy (SEM), we evaluate the diameter of the fibers, the porosity of the tissue papers, and the thickness of the paper (Fig. R6). The characters of the papers are summarized in Table R2.

Fig. R6 The microstructure of different papers by the scanning electron microscopy (SEM). **a**, Xuanzhi. **b**, Tempo. **c**, Grazie. **d**, Elleair. Scale bar, 100 μm .

Table R2 The structural features of different kinds of paper.

Brand	Porosity (%)	Diameter of fibers (μm)	Thickness (μm)	Morphology of fibers
Xuanzhi	7.66	5~12	140	Tubular, entangled
Tempo	4.80	10~40	215	Flat, parallel
Grazie	6.12	10~25	140	Flat, crimped
Elleair	11.7	9~25	115	Flat, parallel

Based on the measured characters, it is concluded that the porosity and the thickness of the tissue show little correlation with the output performance of the HHC device, while the diameter and the morphology of the paper fibers show influences on the performance. The paper with slender and tubular fibers, such as Xuanzhi, can generate the largest open-circuit voltage, as more complex networks can be formed in the tissue.

Manuscript revision:

Fig. R5 and Fig. R6 are added to the Supplementary Information as Fig. S14.

Comments for Reviewer #3:

Beyond that, we asked the reviewers to look over your responses to Reviewer #3 and while they believe the revision is satisfactory, they recommend elaborating further on the efficiency of this device in the main text.

Reply: We thank the reviewers for the advice. We agree that the discussion on the efficiency of the device should be mentioned to the main text, to address the reader's concerns on our output performance.

Manuscript revision:

The discussion on the efficiency of the HHC is added to the Manuscript, Page 9.

Reviewer #1 (Remarks to the Author):

The authors have addressed my comments and the manuscript can now be published.

Reply: We thank the reviewer for the kind help on reviewing our manuscript.

Reviewer #2 (Remarks to the Author):

After thoroughly reviewing the comments and revised manuscript, the author has answered most of the questions. However, I suggest that the author provide efficiency and provide a detailed explanation of the efficiency of this device in the main text, which is very important from an application perspective as this aspect has rarely been addressed in previous publications.

Reply: We sincerely thank the reviewer for re-reviewing our manuscript and providing the suggestion that moving the discussion of the efficiency in the main text, which we believe will greatly enhance the impact of our work. Based on the comments, we extend the discussion of the efficiency calculation (page 9, lines 172-179 in our Revision round 2) as a new paragraph in the main text.

Manuscript revision:

Page 9-10, lines 176-192, added the new paragraph to discuss the energy conversion efficiency calculation and comparison of the hydrovoltaic devices.